# Locally collective hydrogen bonding isolates lead octahedra for white emission improvement

Bin-Bin Cui [1,2,10]*, Ying Han[1,2,3,10], Bolong Huang [4]*, Yizhou Zhao[3], Xianxin Wu [5,6], Lang Liu[3], Guangyue Cao[1,3], Qin Du[3], Na Liu[3], Wei Zou[7], Mingzi Sun [4], Lin Wang[8], Xinfeng Liu [5], Jianpu Wang [7], Huanping Zhou [9] & Qi Chen [1,3]*

As one of next-generation semiconductors, hybrid halide perovskites with tailorable optoelectronic properties are promising for photovoltaics, lighting, and displaying. This tunability lies on variable crystal structures, wherein the spatial arrangement of halide octahedra is essential to determine the assembly behavior and materials properties. Herein, we report to manipulate their assembling behavior and crystal dimensionality by locally collective hydrogen bonding effects. Specifically, a unique urea-amide cation is employed to form corrugated 1D crystals by interacting with bromide atoms in lead octahedra via multiple hydrogen bonds. Further tuning the stoichiometry, cations are bonded with water molecules to create a larger spacer that isolates individual lead bromide octahedra. It leads to zero-dimension (0D) single crystals, which exhibit broadband 'warm' white emission with photoluminescence quantum efficiency 5 times higher than 1D counterpart. This work suggests a feasible strategy to modulate the connectivity of octahedra and consequent crystal dimensionality for the enhancement of their optoelectronic properties.

[1] Advanced Research Institute of Multidisciplinary Science, Beijing Institute of Technology (BIT), Beijing 100081, P. R. China. [2] School of Chemistry and Chemical Engineering, BIT, Beijing 102488, P. R. China. [3] Beijing Key Laboratory of Construction Tailorable Advanced Functional Materials and Green Applications, Experimental Center for Advanced Materials, School of Materials Science and Engineering, BIT, Beijing 100081, P. R. China. [4] Department of Applied Biology and Chemical Technology, The Hong Kong Polytechnic University Hung Hom, Kowloon, Hong Kong, P. R. China. [5] CAS Key Laboratory of Standardization and Measurement for Nanotechnology, CAS Center for Excellence in Nanoscience, National Center for Nanoscience and Technology, Beijing 100190, China. [6] University of Chinese Academy of Sciences, Beijing 100049, China. [7] Key Laboratory of Flexible Electronics & Institute of Advanced Materials, Jiangsu National Synergetic Innovation Center for Advanced Materials, Nanjing Tech University, Nanjing 211816, P. R. China. [8] School of Mechatronical Engineering, BIT, Beijing 102488, P. R. China. [9] Department of Materials Science and Engineering, College of Engineering, Peking University, Beijing 100871, P. R. China. [10] The authors contributed equally: Bin-Bin Cui, Ying Han. *email: cui-chem@bit.edu.cn; bhuang@polyu.edu.hk; qic@bit.edu.cn

As burgeoning materials for optoelectronic and photonic devices, metal-halide perovskites have attracted much attention for their changeable crystal structures, tunable semiconductor properties and quantum confinement effect in low-dimensionality recently[1,2]. Latest discovered zero-dimensional (0D), one-dimensional (1D) and two-dimensional (2D) halide perovskites are promising phosphors for efficient illumination[3,4], wherein the quantum confinement effect at molecular levels, together with exciton self-trappings states, lead to excellent photoluminescence (PL) and unique applications potentially. When constructing perovskite materials, compatible cations are introduced into the framework composed of metal-halide octahedra to tune their connecting modes, which results in the variation in crystal structures and emission properties in this family of materials[5,6]. Especially, radiative transition from self-trapping levels to ground states (electron-phonon coupling) results in large Stokes shifts and broadband emission of these materials[7,8].

While most halide perovskites prefer to expose (100) crystal planes, (110)-oriented corrugated 2D hybrid halide perovskites are rare and yet less exploited. They show unique connectivity of lead halide octahedra with broadband white-emission, which is appealing for white-light emitting diodes (WLEDs)[9,10]. For example, Karunadasa et. al recently reported two 2D Pb-Br perovskites, (N-MEDA)[PbBr$_4$] (N-MEDA = $N^1$-methylethane-1, 2-diammonium) and (EDBE)[PbBr$_4$] (EDBE = 2, 2′-(ethylene-dioxy)bis(ethylammonium)), which both emitted 'warm' white-light with stable PLQEs up to 9%[11,12]. The broadband white-emission is ascribed to the strong electron-phonon coupling in self-trapped excitons (STE) due to transmutable lattices and inhomogeneous intrinsic trap states. Only recently, the sliced 2D emitters reported by Gautier et al. have the inorganic framework chain composed of edge-sharing lead halide octahedra, which improves the PLQE significantly possibly due to the deformable post-perovskite-type chains with weak organic-inorganic interactions[13].

In addition to the 2D layout, perovskite crystals are often constructed by corner-shared, edge-shared, and face-shared metal-halide octahedral wires (1D), and even individual metal-halide octahedron (0D) that is isolated by specific organic cations with 'core-shell' configurations[14,15]. Owing to abundant exciton self-trapping states and strong electron-phonon coupling in 'host-guest' perovskites, 1D and 0D perovskites with strong quantum confinement show highly efficient PL[16]. For example, Ma et al. reported efficient bluish white-emission in 1D organic lead bromide perovskites (C$_4$N$_2$H$_{14}$PbBr$_4$) with a PLQE up to 20%[17], and 0D tin bromide based hybrids (C$_4$N$_2$H$_{14}$Br)$_4$SnBr$_6$ with an emission in the yellow region[18–20]. Zhou et al. achieved efficient white emission of 1D perovskite crystals by doping Mn$^{2+}$ ions. The highest PLQE value is approximately 28% for the Mn-doped 1D perovskites which containing 1.3% Mn$^{2+}$ doping, representing a significant improvement over that of the intrinsic 1D perovskites of approximately 12% with a blue light[6].

Apart from 0D tin based organic metal halide hybrids, 0D lead-based perovskites are recently revealed in all-inorganic Cs$_4$PbBr$_6$ single crystals and nanocrystals[21,22]. Interestingly, they were observed to show outstanding narrow green luminescence[18], which is quite different from that of its Sn counterpart. It leads to an intense debate with respect to the underlying PL mechanism. The green luminescence in Cs$_4$PbBr$_6$ perovskite is proposed to be stemmed from the defect state of bromide vacancies (V$_{Br}$) possibly, and/or from CsPbBr$_3$ impurities within host crystals[23]. Further investigations show that Cs$_4$PbBr$_6$ emits no obvious visible light at room temperature, but it exhibits the emission of 375 nm at 4.2 K[24]. Unfortunately, these evidences are not strong enough to confirm the origin of green emission in 0D lead halide

perovskite. Here we report the organic-inorganic 0D hybrid perovskites-like materials composed of individual isolated [PbBr$_6^{4-}$] octahedra, which exhibits the optical properties completely different from that of the all-inorganic Cs$_4$PbBr$_6$ 0D perovskites.

We synthesize N-(Aminocarbonyl)-1,2-diaminoethane hydrobromide (C$_3$N$_3$H$_9$O·2HBr) (Fig. 1a) with strong intermolecular hydrogen bonds to construct low-dimensional organic lead bromide hybrids. By tuning the precursor stoichiometry, the amides interact with local water molecule via multiple hydrogen bonding to create larger spacers, which isolate [PbBr$_6^{4-}$] blocks to reduce materials dimensionality from 1D to 0D (Fig. 1b, c). Upon excitation, broadband 'warm' white and large stokes shift emission is realized in the 1D crystal and the 0D organic lead bromide hybrid composed of individual [PbBr$_6^{4-}$] species, wherein the PLQE improves over 5 times by regulating the dimensionality from 1D to 0D. Thus, we have not only achieved the controllable growth of single crystals in different dimensionalities simply by merely changing the precursor stoichiometry, but also demonstrated the intrinsic PL from quasi-individual [PbBr$_6^{4-}$] in hybrids materials. Locally collective hydrogen bonds within organic cations and water provide extra possibility to create metal halide hybrids with desirable inorganic framework structures, which can be extended to other hybrid materials systems.

## Results

**Materials synthesis.** Urea-amide hydrobromide, N-(Amino-carbonyl)-1,2-diaminoethane hydrobromide (C$_3$H$_9$N$_3$O·2HBr)[25], was synthesized as organic ligands to build our low-dimensional lead bromide hybrids. Pure white solid urea-amide hydrobromide in 65% yield was recrystallized from the crude product. As shown in Fig. 1a, when the ratio of precursory C$_3$N$_3$H$_9$O·2HBr and PbO is 3: 2, 1D molecular assemblies are formed, in which the corrugated 1D hybrid corner-sharing octahedral lead bromide [Pb$_2$Br$_9^{5-}$] are cleaved out by two types of urea-amide cations C$_3$N$_3$H$_{10}$O$^+$ and C$_3$N$_3$H$_{11}$O$^{2+}$. Special 'chelating effect' of intermolecular hydrogen bonding stabilized its corrugated-liner structure. Interestingly, with the ratio of C$_3$N$_3$H$_9$O·2HBr:PbO increasing to 2:1, urea-amides interact with water molecules through collective hydrogen bonding to create a larger molecular spacer, which can separate individual [PbBr$_6^{4-}$] octahedron to construct 0D (C$_3$N$_3$H$_{11}$O)$_2$PbBr$_6$·4H$_2$O. Yields of these two crystals are around 55 to 62%. C$_3$H$_9$N$_3$O·2HBr is characterized by proton nuclear magnetic resonance ($^1$H NMR) (Supplementary Fig. 1). $^1$H NMR (DMSO-d6): δ 2.80 (q, $J = 6.0$ Hz, 2H, CH$_2$NH$_3^+$), 3.20 (t, $J = 6.0$ Hz, 2H, CONHCH$_2$), 5.72 (s, 3H, CONH$_3^+$), 6.22 (s, 1H, CONHCH$_2$), 7.71 (s, 3H, $^+$NH$_3$). Therefore, two acid H$^+$ belong to two terminal amines in C$_3$H$_9$N$_3$O·2HBr, respectively. Single crystal X-ray diffraction (SCXRD), infrared spectroscopy (IR), powder X-ray diffraction (PXRD) and thermogravimetric analysis (TGA) represent the composition of bulk 1D or 0D crystals comprehensively. Details of the synthesis and characterization methods are shown in the Methods.

**Structure characterization.** As shown in Fig. 1, SCXRD determined crystal structures of 1D and 0D organic metal bromide hybrids. Specific crystal cell parameters are summarized in Supplementary Table 1. In particular, urea-amide cations in 1D (C$_3$N$_3$H$_{10}$O)(C$_3$N$_3$H$_{11}$O)$_2$Pb$_2$Br$_9$ are protonated in two states: one-valence C$_3$N$_3$H$_{10}$O$^+$ and two valence C$_3$N$_3$H$_{11}$O$^{2+}$ (Fig. 1d). As such, this material adopts a monoclinic space group *P21/n* with corrugated 1D chains surrounded by C$_3$N$_3$H$_{10}$O$^+$ and C$_3$N$_3$H$_{11}$O$^{2+}$ cations. The 1D structure derives from the (110)-oriented corrugated 2D perovskites by slicing the inorganic layers,

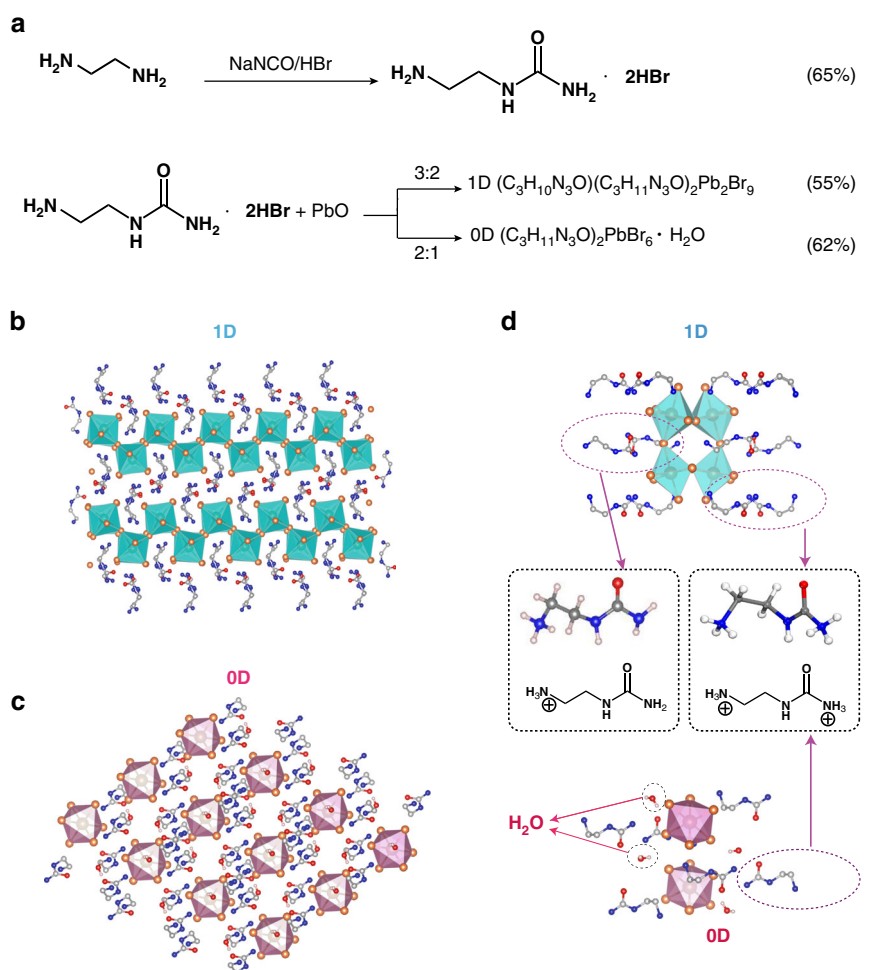

**Fig. 1** The synthesis and structural characterization of crystals of 1D and 0D lead bromide hybrids. **a** Schemes for the synthesis of urea-amide hydrobromide and 1D, 0D lead bromide perovskite single crystals; Crystal structures of **b** 1D $(C_3N_3H_{10}O)(C_3N_3H_{11}O)_2Pb_2Br_9$ and **c** 0D $(C_3N_3H_{11}O)_2PbBr_6·4H_2O$, respectively (brown spheres: lead atoms; orange spheres: bromine atoms; blue spheres: nitrogen atoms; gray spheres: carbon atoms; indigo and pink octahedron: $[PbBr_6^{4-}]$; hydrogen atoms are hidden for clarity). **d** Spatial configuration of $C_3N_3H_{11}O^{2+}$ and $C_3N_3H_{10}O^+$ in 1D and 0D crystals

which are connected by only two rows of corner-sharing $[Pb_2Br_9^{5-}]$ octahedra. PXRD pattern of the ball-milled samples are identical to the simulated PXRD pattern from the single crystals (Supplementary Fig. 2), further confirming the structures of 1D $(C_3N_3H_{10}O)(C_3N_3H_{11}O)_2Pb_2Br_9$ and 0D $(C_3N_3H_{11}O)_2PbBr_6·4H_2O$ hybrids. The formation of wavy inorganic assemblies is due to the steric effects of cation inserted (Fig. 1d and Supplementary Fig. 3). It is impossible to further extend a 2D layer along the (110) direction, and 1D nanowires are thus cleaved out.

The major difference in the 0D and 1D organic lead bromide hybrid is existence of hydrogen bonding. As shown in Fig. 2a, there are no significant intermolecular interactions between urea-amide cations in 1D materials. In 0D structure, the unique cation provides abundant sites to form locally collective hydrogen bonding with adjacent molecules successfully separating the individual $[PbBr_6^{4-}]$. The cation possesses the primary amine group (marked 1 in Fig. 2b), secondary amide group (marked 2 in Fig. 2b), and the primary amide group -CO-NH$_3$ (marked 3& 4 in Fig. 2b), which are essential to construct the 0D crystal. The combination of the amine/amide groups (1 & 2) with special configuration effectively interact with bromide atom in lead octahedra via multiple hydrogen bonds (Supplementary Fig. 4a, highlighted in blue), resulting in a corrugated configuration. To

further cut out the octahedra individually, the functional group -CO-NH$_3$ (marked 3 & 4 in Fig. 2b) at the end of the cation takes effects, which provides abundant sites to form locally collective hydrogen bonding with adjacent molecules. When increasing the concentration of the cation, it has strong intermolecular hydrogen bonding with H$_2$O that creates a large spacer to isolate the individual lead bromide octahedra to form a 0D structure. Therefore, the local hydrogen bonding collectively contributes to tune the connectivity of lead octahedra, leading to lead halide hybrids with different dimensionality. We summarized the hydrogen bonding taking effects in three categories labelled in different colors and measured their bonding lengths in 0D crystal (Fig. 2c and Table 1): (1) water/amine group (blue) N–H···O–H (2.097, 2.347, 2.812, 2.934 and 2.982 Å), (2) water/carboxyl group (red), H–O–H···O=C (2.239, 2.964 Å) and (3) water/water (green) H–O–H···O–H (2.172 Å). Detailed hydrogen bonding parameters including bond lengths and angles for cations in 0D crystals are marked in Supplementary Fig. 4b. The calculated hydrogen bond strength for 0D lead bromide hybrids is listed in Supplementary Table 2. To be noted, SCXRD measurement cannot directly tell the proton positions over the organic cation. Such assignment leads to the lowest energy of the entire cation predicted by DFT calculation (Fig. 2a, b and Supplementary Tables 3–6).

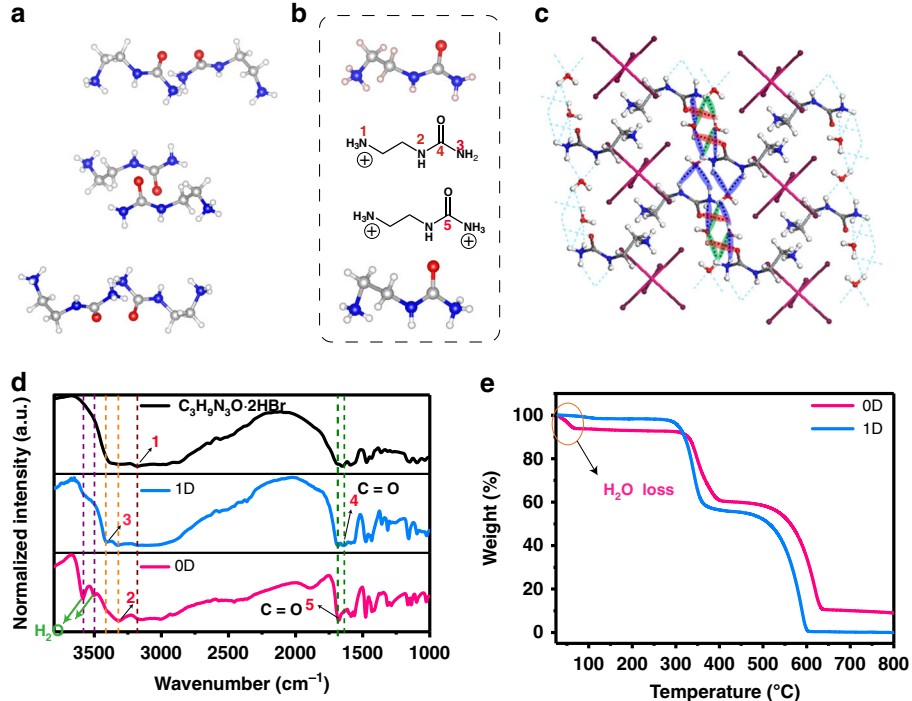

**Fig. 2** Characterization of hydrogen bonds in crystals of 1D and 0D lead bromide hybrids. **a** Organic cations in the 1D lead bromide crystal. **b** Spatial configuration of $C_3N_3H_{11}O^{2+}$ and $C_3N_3H_{10}O^+$ with functional groups being marked. **c** Highlights of the hydrogen bonds between organic cations and water in the 0D lead bromide crystal. **d** Infrared spectroscopy (IR) analysis of abrasive 1D and 0D crystals. **e** Thermogravimetric analysis (TGA) for weight loss curves of 1D and 0D crystals

| Table 1 Hydrogen Bond Parameters for 0D lead bromide hybrids | | |
|---|---|---|
| **D−H⋯A** | **D⋯A (Å)** | **angle at H (deg)** |
| H-O-H⋯O=C | 2.239 | 100.478 |
| N-H⋯O-H | 2.964 | 107.777 |
| N-H₃⋯O-H | 2.347 | 123.268 |
| H-O-H⋯O-H | 2.097 | 164.159 |
| | 2.812 | 110.452 |
| | 2.934 | 162.49 |
| | 2.982 | 126.064 |
| | 2.172 | 149.31 |
| D, H-donor; A, H-acceptor | | |

To further confirm there are two kinds of urea-amide cations in 1D $(C_3N_3H_{10}O)(C_3N_3H_{11}O)_2Pb_2Br_9$, IR of urea-amide and two crystals were conducted (Fig. 2d). Vibrational signals of $H_2O$ in 0D $(C_3N_3H_{10}O)_2PbBr_6\cdot4H_2O$, two different carbonyl group signals and three kinds of amide group signals in 1D $(C_3N_3H_{10}O)(C_3N_3H_{11}O)_2Pb_2Br_9$ were detected by IR. Compared to 1D system however, amide signal of 3409 cm⁻¹ (marked as 3 in Fig. 2d) and carbonyl signal of 1636 cm⁻¹ (marked as 4) do not appear in 0D $(C_3N_3H_{11}O)_2PbBr_6\cdot4H_2O$. It is likely attributed to hydrogen bonding of $H_2O$ molecules, which demonstrates the constitutions' difference of $C_3N_3H_{10}O^+$ and $C_3N_3H_{11}O^{2+}$ cations. TGA results also proved the existence of water molecule in 0D $(C_3N_3H_{11}O)_2PbBr_6\cdot4H_2O$ (Fig. 2e), wherein a weight loss of 5% was observed at 70 °C.

**Photoluminescence properties**. Colorless crystals of 1D and 0D lead bromide hybrids under ambient light show yellowish-white emissions upon 365 nm irradiation, wherein 0D bulk crystal is

brighter than 1D bulk crystal (Fig. 3a, b). As shown in Fig. 3c, 0D $(C_3N_3H_{11}O)_2PbBr_6\cdot4H_2O$ can be excited by UV from 280 to 400 nm producing an emission peak at 568 nm while 1D $(C_3N_3H_{10}O)$ $(C_3N_3H_{11}O)_2Pb_2Br_9$ can be excited by UV from 300 to 420 nm producing an emission peak at 530 nm. The absorption spectra of the 0D crystals are in good agreement with their excitation spectra (Supplementary Fig. 5a). Both 1D and 0D lead bromide hybrids can emit a broadband emission range from 400 to 850 nm with a wide FWHM of 160 and 200 nm, respectively, and the Stokes shifts for bulk 1D and 0D lead bromide hybrids are 141 and 218 nm, respectively. The luminescence decay of emission for bulk 1D and 0D crystals are 14.2 and 15.4 ns (Fig. 3d), respectively, which follow the pattern of short single exponential decay. The Commission Internationale de l'Eclairage (CIE) chromaticity coordinates of these yellowish-white emissions are (0.39, 0.42) and (0.42, 0.44) for 0D and 1D crystals (Fig. 3e), respectively. Compared with a pure white light (0.33, 0.33), 0D and 1D lead bromide hybrids in this work emit 'warm' white light that is suitable for indoor illumination[26,27]. Both bulk 1D and 0D crystals retained emission color and intensity during one month continuous irradiation under a 365 nm in ambient environment. The broad emission of 0D bulk crystal becomes intense and narrow with decreasing temperature (Fig. 3f and Supplementary Fig. 5b), indicating the exciton self-trapping with less vibrational relaxation.

To confirm it is the STE mechanism, we conducted the transient absorption (TA) measurement. As is shown in Fig. 3g, by the above-excitonic-peak excitation at 348 nm with low pulse energy of 14 μJ cm⁻² per pulse, a broad pump-induced absorption with lower energy than that of free excitation state was observed in our 0D crystals. In TA spectra of 2D and 3D lead-iodide hybrid perovskites with narrow emission, it often exhibits below-exciton bleaching features owing to filling of permanent trap states[28]. The distinguished TA spectra suggest the wide emission is probably assigned to STE[29], which

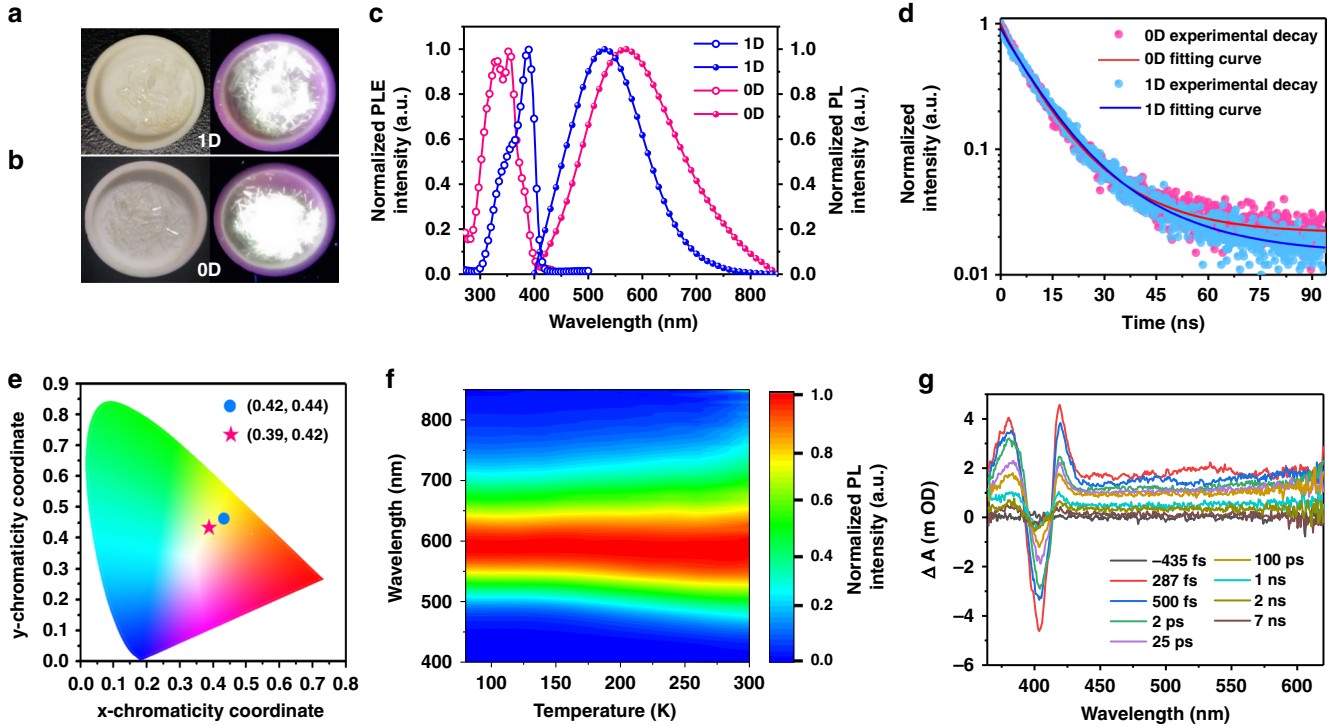

**Fig. 3** Photophysical properties of 0D and 1D lead bromide perovskites. **a**, **b** Photos of 1D and 0D crystals in ambient and under UV light (365 nm), respectively. **c** Excitation (open circle) and emission (circle) spectra of 1D (blue) and 0D (pink) crystals at room temperature. **d** Time-resolved PL experimental decay (circle) and fitting curve of the 0D (red line) and 1D bulk crystals (blue line) probed at 568 nm and 530 nm, respectively. **e** CIE chromaticity coordinates of 1D (dot) and 0D (star) crystals. **f** Temperature dependent emission spectra of 0D ($C_3H_{11}N_3O)_2PbBr_6\cdot4H_2O$ crystals. **g** Transient absorption spectrum of 0D crystals upon photoexcitation at 348 nm

is stemmed from the instantaneous STE energy level produced during the excitation process[1]. Upon photoexcitation, the as-formed excitons quickly relax to multiple self-trapped states with different energies, which leads to a white broadband emission. The broadband emission intensity shows a linear dependence on the power density of excitation up to 500 W cm$^{-2}$ (Supplementary Fig. 5c), suggesting that emission is irrelevant to the intrinsic defects. Moreover, the emission of bulk crystals and ball-milled powder are identical (Supplementary Fig. 6b), indicating that emission does not origin from surface defects. The emission characteristics are almost independent of different excitation wavelengths (Supplementary Fig. 6d). The STE mechanism proposed in low-dimensional hybrid perovskites is different from conventional organometal emitters (e.g. Iridium complex)[30], which involves metal to ligand charge transfer (MLCT) that excitons are mostly localized within either specie. We checked the PL of the organic cation, the emission spectrum of $C_3H_9N_3O\cdot2HBr$ salts powder lies in the blue region with the peak at around 393 nm (Supplementary Fig. 7). It is different from the broadband emission of our 0D single crystal, suggesting the PL is not stemmed from the urea hydrobromide[31,32].

To exclude the possible emission of 0D crystals origin from the defects, we also calculated the dominant vacancy defects $V_{Br}$ (3.0 eV) and $V_{Pb}$ (4.2 eV). Neither of transition levels of $V_{Br}$ and $V_{Pb}$ matches well with the broadband emission at 568 nm (2.18 eV) (Supplementary Fig. 8a, b), further suggesting that the emission origin from the self-trapped states rather than intrinsic vacancy defects. From temperature-dependent PL measurements (Supplementary Fig. 9), the exciton binding energy are estimated to be 141 and 124 meV for 0D and 1D perovskites, respectively[21]. The PLQE of bulk 0D crystal is measured to be 9.6%, which is five times that of the 1D counterpart (1.7%).

## Discussion

We tend to understand the photophysical behavior by DFT calculations on the electronic configurations of the two materials of interest. The electronic structures of the 0D and 1D lead bromide hybrids regarding the band structures and electronic properties are illustrated. The 0D and 1D systems show direct bandgap about 3.74 and 3.02 eV, respectively (Fig. 4a, b). The projected density of states (PDOS) shows slightly different results of the two systems. For 0D crystals, the valence band maximum (VBM) and conduction band minimum (CBM) are mainly contributed by Br-4p and Pb-6p, which resembles the collective contributions of HOMO-LUMO levels of periodic 0D distributions. It agrees with the other lead halide perovskites with different dimensionalities[15]. However, the contribution of the organic spacer is also noted, especially at the CBM (Fig. 4c). On contrast, the CBM of 1D system is dominated by Pb-6p. Meanwhile, the contribution of the organic spacer in VBM becomes evident (Fig. 4d). In both 0D and 1D systems, no extra states within the bandgaps induced by organic spacer are found. Norm-conserving pseudopotential to the treatment supplies highly similar band structures of 0D and 1D systems due to the subtle bandgap size changes (Fig. 4e, f). The 3D spatial orbital distribution also supports the contribution of organic spacer in the electronic structures. In 0D system, the electronic states are localized near $[PbBr_6]^{4-}$ octahedra while the hole states are mainly localized within the organic spacer region (Fig. 4g). On the contrary, the organic spacer mainly contributes to electronic states and the Pb-6p anti-bonding states dominate the delocalized hole state levels (Fig. 4h). The corresponding traps in terms of PDOSs analysis and hydrogen bonding strength calculations are all explained in the Supplementary Fig. 8 and Supplementary Note 1.

Combining with the PL results, DFT calculations on band structures suggest the emission of 1D bulk crystal is originated

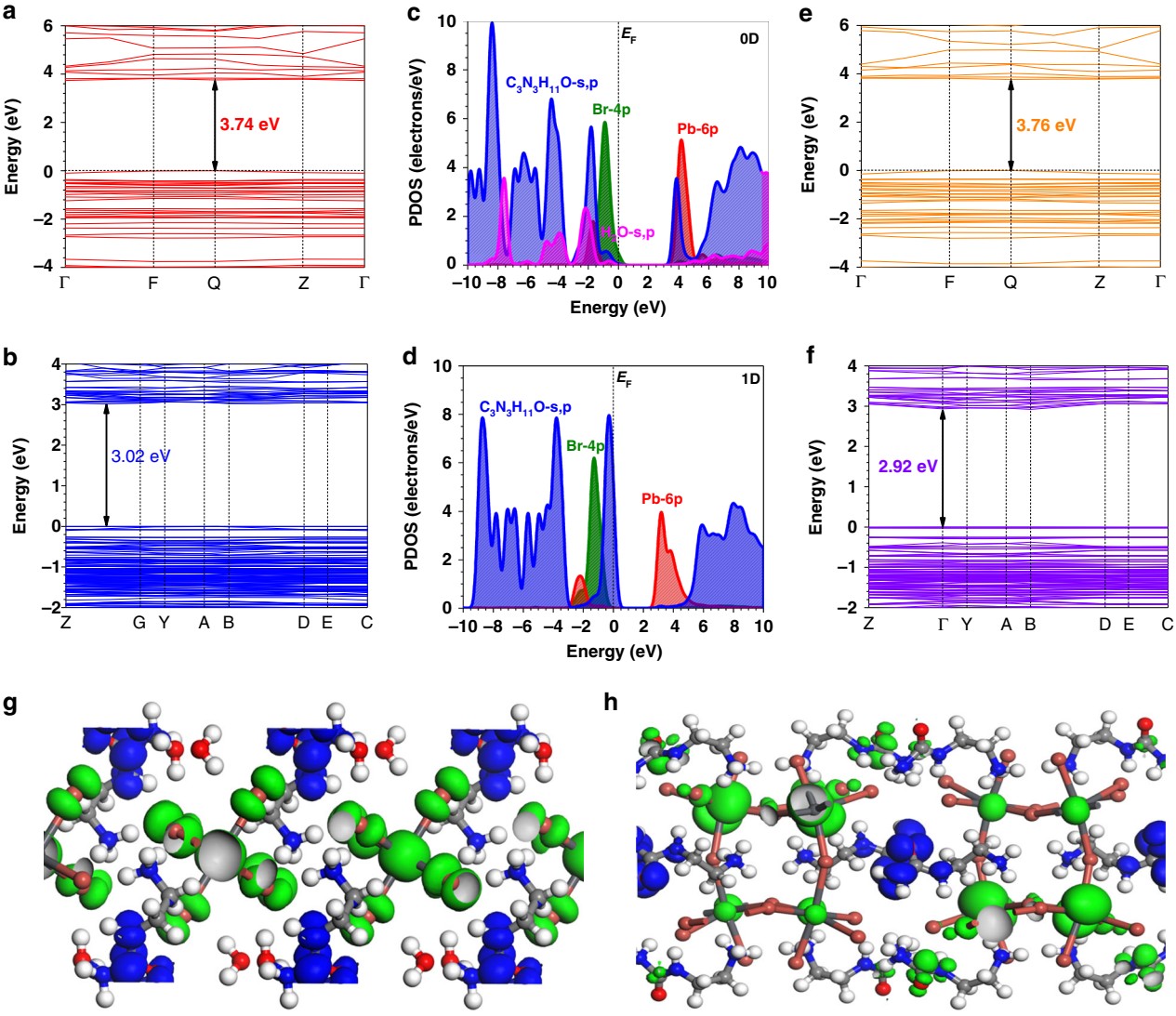

**Fig. 4** Density functional theory (DFT) calculations on band structures and photoluminescence mechanisms. **a, b** Calculated band structure of 0D and 1D lead bromide hybrids under ultrasoft pseudopotential. Inset: Zoom-in valence band (VB) structure. **c, d** Projected density of states of 0D and 1D lead bromide hybrids, respectively. **e, f** Calculated band structure of 0D and 1D lead bromide hybrids under norm conserving pseudopotential. Inset: Zoom-in valence band (VB) structure. **g, h** Real spatial contour plots for bonding and anti-bonding orbitals near $E_F$ for 0D and 1D lead bromide hybrids, respectively

from self-trapped excited states derived from clusters assembled by $[PbBr_6]^{4-}$ octahedra[17]. With respect to that of 0D crystals, it is generally explained by molecular excited-state structural reorganization[18]. However, when we check the transient PL lifetimes, our 0D crystals possess the exciton lifetime of 15.4 ns, which resembles that of 1D counterpart (14.2 ns) rather than microsecond lifetime. (Fig. 3d) The results deviate from that of tin based crystals ever reported[14,18], which indicates the possible photophysical behavior for our 0D crystals is more likely to involve the self-trapped states. Comparing to tin based crystals wherein tin halide octahedra behave more individually, the spatially isolated lead halide octahedra are more prone to interfere with each other electronically. It is reasonable since lead atom shows larger size and higher density of inner electrons, wherein the outer electrons are more dispersive spatially as compared to the tin atom. Moreover, organic cations in our crystals show a reasonable extent of energy states involvement, which can reinforce the interaction between individual lead halide octahedra. Detailed mechanisms are under investigations now.

In addition, it is noticed that our 0D crystals shows higher PLQE than the 1D counterpart. Considering the excess $H_2O$ in

lattice may influence the luminescence property of 0D crystal, we have carried out on controlled experiments for comparison. The measured PLQE for acetone washed only and 24 h vacuum-dried (at 90 °C) 0D crystals are 9.3% and 2.4%, respectively. $H_2O$ loss indeed decreased the luminescence property. However, after dried the 0D crystal under vacuum at 90 °C for 24 h, most 0D single crystals were transformed to 1D phase, which was verified by PXRD (Supplementary Fig. 10). In addition, PL spectrum (Supplementary Fig. 11) for the vacuum-dried sample shows the broadband emission of 0D with an extra emission peak of $PbBr_2$. It further confirms the change of crystal structure which leads to the reduction in PLQE. Mainly we refer to DFT simulation to explore possible mechanisms that may explain the increased PLQE. Based on the DFT calculations, the isolated $[PbBr_6^{4-}]$ octahedra do not interact with each other strongly due to the neighboring larger spacer, the Br 4p and Pb 6p derived bands are quite flat and almost dispersionless. It indicates lowering the dimensionality from 1D to 0D promotes the localization of excitons, as a result, it is not favored for resonant energy transfer in 0D crystals, which shows lower probability of trapping at the intrinsic defects and nonradiative

recombination during exciton migration. In short, the resulting immobile excitons in our 0D decrease the non-radiative recombination[16], which possibly explains the increased PLQE as observed in experiment.

In addition, dielectric constant correlates to the polaron behavior, which also affects the emission properties. In order to measure the dielectric constant in 0D and 1D crystals, we fabricated the sheet samples with the thickness about 1 mm and diameter 1 cm by cold pressing, and dielectric measurements were performed in the frequency range 50 Hz to 20 MHz (Supplementary Fig. 12). The real part of dielectric constants of 0D and 1D crystals are 11.75 and 11.27 at about 1 MHz, respectively, which are much smaller than the 3D halide perovskite (often over 50)[33,34]. The measured dielectric constant of our 0D crystals is similar to other reported 0D halide perovskite-like crystals (Supplementary Table 7)[16]. Therefore, the presence of hydrogen bonding in the 0D structure may not produce large polarons as in conventional 3D perovskites. It induces small polarons by strong electron-phonon coupling and carrier trapping, which leads to STEs.

To wrap up, a powerful toolbox is demonstrated to tune the connecting modes of lead halide octahedra in perovskite crystals and consequent materials properties via locally collective hydrogen bonding effects. We employ a unique urea-amide cation interacting with bromide atoms in lead octahedra via multiple hydrogen bonds to form a corrugated 1D structure. Interestingly, when increasing the cation ratios, it provides abundant sites to form collective hydrogen bonding with local $H_2O$ molecules that create a large spacer to isolate the lead bromide octahedra to construct a 0D crystal. Upon excitation, both two crystals emit broadband 'warm' white light, wherein the PLQE improves over 5 times when regulating the dimensionality from 1D to 0D. More importantly, locally collective hydrogen bonding effects between organic molecules like urea-amide cations can be employed to explore organic-inorganic hybrids with different connecting modes in inorganic framework, and further guide the design of other hybrid materials systems beyond lead halide hybrids.

## Methods

**Materials**. Lead (II) oxide (PbO, ≥99.9%), 1,2-diaminoethane (≥99.0%), sodium cyanate (≥97.0%), hydrobromic acid (48 wt.% in $H_2O$), acetone (HPLC grade) and diethyl ether (analytical reagent) were purchased from Aladdin Industrial Corporation (Shanghai). All ingredients and solvents were used without further purification unless otherwise stated.

**Preparation of urea-amide hydrobromide**. To a round-bottom flask add (0.05 mol, 3.01 g) 1,2-diaminoethane, (0.06 mol, 4.08 g) sodium cyanate, and then 25 ml $H_2O$, hydrobromic acid (48 wt.% in $H_2O$) were injected. Stirring the mixture for 10 min at room temperature (R. T.) resulted a clear solution. A white solid residue was obtained after the solution heated at 60 °C for 12 h and evaporated under vacuum. Redissolved the solid residue in 25 ml $H_2O$ with 5 mL hydrobromic acid solution (48 wt.% in $H_2O$), and then evaporated the liquid again. Finally, 5.95 g pure white solide N-(Aminocarbonyl)-1,2-diaminoethane hydrobromide ($C_3H_9N_3O \cdot 2HBr$) in 65% yield was filtered out after recrystallized the solid residue in ethanol/diethyl ether for three times.

**Growth of corrugated 1D $(C_3H_{10}N_3O)(C_3H_{11}N_3O)_2Pb_2Br_9$ bulk crystals**. PbO (0.90 mmol, 200.0 mg) powder was dissolved in a mixture of 4 mL 48 wt.% aqueous HBr solution by heating 90 °C under constant magnetic stirring for about 5 min, forming a clear yellow solution. Subsequent addition of (1.35 mmol, 248.4 mg) solid $C_3H_9N_3O \cdot HBr$ to the hot solution. The stirring stopped, and the solution was left to cool to R. T. and set up in vapor diffusion chambers with diethyl ether. Colorless and needle-shaped crystals of 1D $(C_3H_{10}N_3O)(C_3H_{11}N_3O)_2Pb_2Br_9$ were obtained through diffusion of diethyl ether into this solution over 24 h in a moderate yield (about 55%).

**Growth of 0D $(C_3N_3H_{10}O)_2PbBr_6 \cdot 4H_2O$ bulk crystals**. PbO powder (0.45 mmol, 100 mg) was dissolved in a mixture of 4 mL 48 wt.% aqueous HBr solution by heating 90 °C under constant magnetic stirring for about 5 min, forming a clear yellow solution. Subsequent addition of (0.9 mmol, 165.6 mg) solid $C_3H_9N_3O \cdot HBr$, stirring until dissolved to the hot yellow solution, and then the solution was cooled down to R.T. and set up in vapor diffusion chambers with diethyl ether. Colorless and needle-shaped crystals of 0D $(C_3N_3H_{10}O)_2PbBr_6 \cdot 4H_2O$ were obtained through diffusion of diethyl ether into this solution over 24 h in a moderate yield (about 62%).

**Physical measurements**. $^1H$ NMR spectra were measured using a Bruker AVANCE III 300 MHz NMR Spectrometer in designated deuterated solvent. TGA was recorded from R.T. to 800 °C with the 10 °C min⁻¹ in nitrogen atmosphere on a TA Instruments TGAQ500. The X-ray diffraction data were performed on a Rigaku Saturn 724 diffractometer with rotating anode (Mo-Kα radiation, 0.71073 Å). The structure data was solved using SHELXS-97 by the direct method and refined using the software of Olex2 and VESTA. CIFs have been deposited with CCDC, and the CCDC No. for 1D and 0D lead bromide crystals are 1894525 and 1894526, respectively. PXRD was performed on a Panalytical X'Pert Powder Diffractometer with a Cu anode (Kα₁ = 1.54060 Å, Kα₂ = 1.54443 Å, Kα₂ /Kα₁ = 0.50000) with an X'Celerator RTMS detector at 298 K. Simulated powder patterns were obtained by vesta software from the SCXRD.

**Optical measurements**. Absorption spectra (with a 1 nm interval) scanning through synchronous of bulk 1D and 0D lead bromide crystals were collected on a FLS980 spectrofluorometer with an integrating sphere at 298 K. Quartz plates were used in spectroscopic measurements. The absolute photoluminescence quantum efficiencies (PLQEs) were acquired on powders by a FLS980 spectrofluorometer coupled to an integrating sphere. Samples were excited by 450 W Xe lamp passed through a single grating Czerny-Turner monochromator. The spectra of the emitted light and any unabsorbed excitation light were measured using a Princeton Instruments Spectra Pro 500i spectrograph fiber-coupled to the sphere. Time-resolved PL measurements were collected using a Time-Correlated Single Photon Counting (TCSPC) on a PicoQuant. Pulverized samples were excited with a pulsed diode laser and detected by a monochromator with a single photon avalanche diode (PDM 100CT SPAD) and processed by a PicoHarp 300 correlating system. PL at different temperatures were measured on a FLS980 spectrofluorometer attached to a OptistatDN cryostat filled with liquid nitrogen. PL intensity dependence on excitation power density measurements were performed on an Edinburgh Instruments PL980-KS transient absorption spectrometer with a continuum Nd: YAG laser. Steady-state PL spectra of the crystals were obtained on a Varian Cary Eclipse Fluorescence spectrophotometer at 298 K.

**Calculations**. We applied the density functional theory (DFT) calculations with CAmbridge Serial Total Energy Package (CASTEP) regarding all the electronic structure calculations[35]. Both ultrasoft and norm-conserving pseudopotentials have been chosen for the comparison of band structures calculations[36,37]. Norm-conserving pseudopotentials of the Pb and Br atoms were generated by the OPIUM code in the Kleinman–Bylander projector form. The electronic exchange and correlation are described by Perdew-Burke-Ernzerhof (PBE) functional within the generalized gradient approximation (GGA)[38]. A nonlinear partial core correction and a scalar relativistic averaging scheme were used to treat the spin–orbit coupling effect. Specifically, we treated (4s, 4p, 4d), (5d, 6s, 6p) as valence states for Br and Pb atoms. The Rappe–Rabe–Kaxiras–Joannopoulos method was chosen to optimize the pseudopotentials for better electronic minimization, particular to denxity mixing scheme using a blocked-Davidson diagonalization method. For all the geometry optimizations, the Hellmann-Feynman forces are set to converged to less than 0.001 eV Å⁻¹ while the total energy has been converged to $5 \times 10^{-5}$ eV per atom. The Monkhorst-Pack grid separation spacing for k-point has been selected with size of 0.07 Å⁻¹ during the process of the geometry optimization based on the Broyden-Fletcher-Goldfarb-Shannon (BFGS) algorithm[39,40].

## Data availability

The data that support the findings of this study are available from the corresponding author upon request.

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

## Acknowledgements
This work was supported by funding from the National Natural Science Foundation (21703008, 51673025) and the Young Talent Thousand Program of China, also funding support from National Key Research and Development Program of China Grant No. 2016YFB0700700. We also thank the financial support from "Cultivate Creative Talents Project" of Beijing Institute of Technology (BIT), and Grant: ZDKT18-01 Open Fund of State Key Laboratory of Explosion Science and Technology (BIT). We highly appreciate the technical discussion with Prof. Jie Ma of BIT and Ms. Xinxin Wang in the context of DFT simulation.

## Author contributions
B.B.C. and Y.H. conceived the low-dimensional perovskite crystals and summarized their measurement data. Y.H. and G.C. synthesized and purified the organic urea-amide hydrobromide; Y.H. synthesized and produced the bulk low-dimensional perovskite crystals and collected Powder XRD, TGA and proton NMR data; Y.H. measured the photophysical and photoluminescence (PL) properties with the help of L.L. and H.Z.; L. W. performed single crystal XRD test; X.W and X.L. performed TA measurement; Y.H. analyzed the SCXRD, TA data with the help of Y.Z., N.L., and Q.D.; W.Z. and J.W. performed PLQY test; M.S. and B.H. were in charge of the part of density functional theory (DFT) calculations. The manuscript was mainly written and revised by Q.C., B.B. C., and Y.H. This project was directed and supervised by B.B.C. and Q.C. All authors discussed the results and commented on the manuscript.

## Competing interests
The authors declare no competing interests.
