## [Peer Review File · Nature Communications]

Reviewers' comments:

Reviewer #1 (Remarks to the Author):

Cui et al. report synthesis and characterization of both 1D and 0D lead-based perovskite-like crystals. The authors identify multiple-hydrogen-bonding as a reasoning behind the creation of a larger molecular spacer, which separates individual [PbBr₆]⁴⁻ octahedron constructing 0D crystals exhibiting broadband warm white emission with 5 times higher quantum efficiency compared to the 1D counterpart. Finally, the authors use their data to confirm that the green emission observed in the Cs₄PbBr₆ is not from individual [PbBr₆]⁴⁻.

The authors do report, to the best of my knowledge, the first 0D organic-inorganic lead-based crystal that will be of interest to many in the field. However, based on the comments below, I find the manuscript still requires critical improvements.

1. The quantum efficiency of this 0D crystals is below other reported white light lead-based 1D and 2D emitters (Zhou et al. ACS Appl Mater Interfaces 2017, 9, 40446-40451 and Gautier et al. Adv. Mater. 2019, 31, e1807383.).

2. Lack of proper explanation regarding the origin of the broadband emission. In this regard, I recommend authors to perform transient absorption measurements.

3. The argument related to the debatable green emission observed in Cs₄PbBr₆ might not be valid. If isolated or individual [PbBr₆]⁴⁻ should exhibit broadband emission and not narrow green emission, why is that broadband emission not observed even in the Cs₄PbBr₆ that do not show green emission? (Zhang et al. Cryst. Growth Des. 2018, 18, 6393-6398 and Akkerman et al. Nano Lett. 2017, 17, 1924-1930)

4. The figures quality is extremely poor except for Figure 4.

Further comments:

1. Page 2 Line 4: identify what "A" site is.

2. Scheme 1a: the 65% under the arrow is confusing and should be moved similar to Scheme 1b.

3. Figure 1: it is very difficult to identify the H₂O molecules.

4. Page 4 Line 45: "C₃H₉N₃O·2HBr is characterized by proton nuclear magnetic resonance (1H NMR) (Supplementary Fig. 1)" Authors should provide some interpretation for the results and not simply report the spectra.

5. Figure 2c: please provide the molecules with numbering of the corresponding atoms or functional groups. What does the y-axis represent in this figure? Importantly, it seems that the carbonyl group (5) exist in the 0D crystal.

6. The 1D crystals appear to be yellowish in color and not colorless. Does that match with absorbance data?

7. Authors should report PLE of both crystals.

8. Figure 3g: what are the "microscale crystals" in the legend? There is no discussion in the manuscript regarding this "microscale crystals".

9. Authors should provide the fitting for the PL decay data.

10. Please check Figures numbers in SI.

11. Page 8 Line 182: "the major emission of 1D bulk". Is there a minor origin of the emission?

12. Would be very interesting if the authors could report the Cl and/or I counterparts.

Reviewer #2 (Remarks to the Author):

The manuscript describes the synthesis, structural and optical characterization of a 0D and a 1D lead halide hybrid systems. In both cases, PbBr₆ perovskite octahedra are present, isolated in the 0D structure, and three corner connected in the 1D case, forming double infinite chains. Hydrogen

bonding is observed, and is suggested as the driving interaction for the formation of these structures. Both phases exhibit white-emission upon UV excitation.

There are several issues that make the manuscript less appealing to a broader audience. The authors are encouraged to seek publication elsewhere. For instance, Scientific Reports may be suitable.

Molecular Scissors: the term implies that the molecules described cut extended PbBr₃ blocks or sheets. However, no such proof is presented. The solution-based synthesis more likely produces the different structures due to charge transfer effects. The 1D structure, with three corners connected, reduces the need for charge balancing, while a larger concentration of organic cations is needed for the 0D structure.

In the 0D structure, small polaron behavior is expected, as in Cs₄PbBr₆. Self-trapping will be similar, and a green emission is expected. In contrast, the presence of hydrogen bonding in the 0D structure likely produces a large local polarizability, with commensurate screening of the small polaron and different emission properties. Measurements of the dielectric constants of both the 0D and 1D phases would be helpful to address the question of the "green" emission for isolated PbBr₆ units. The "solid evidence" of characteristic broad band emission may need additional support.

The DFT calculations are consistent with expectations, with the 0D phase showing a similar band gap as for Cs₄PbBr₆. How was the hydrogen bonding included in the electronic structure calculations?

The 1D structure shows virtually no dispersion in the valence and conduction band, making the structure electronically very similar to the 0D phase.

The fact that the 1D structure contains linear Pb₂Br₉ moieties does not produce a noticeable dispersion along these moieties. The observed optical properties are thus expected to be similar to the 0D phase.

Reviewer #3 (Remarks to the Author):

Chen and co-workers synthesized new 1D and 0D hybrid perovskite with broadband "warm" white emissions using the urea-amide cations. The authors found that the higher PLQY of 0D crystal than 1D case could be attributed to the multiple-hydrogen-bonding (MHB) effects which is reasonable explanation. The authors also carefully conducted several experiments to fully characterize the materials from few angles. Overall, the work is very interesting for the low-dimensional perovskite community; and it does indeed deserve publication in Nature Communications after revisions noted.

1) The authors claimed that the higher PLQY of 0D crystal is attributed to the organic cations assembly and the reduction of the thermal vibrations. Does the excess H₂O also play a role in increasing the PLQY of 0D hybrid crystal?

2) Although the authors mentioned that "Both bulk 1D and 0D crystals retained emission color and intensity for at least one month" Does 0D crystal retain the same crystal structure at the high temperature after H₂O loss? And how about the PL spectra and PLQY of 0D crystal at high temperature after H₂O loss? This needs to be clarified in the revised version of the manuscript.

3) I strongly recommend the authors to measure the exciton binding energy of the 0D structure and compared it to the inorganic 0D perovskite structure. This is also beneficial to take this new materials to the next level in terms of application.

4) In Fig 4, it is difficult for the readers to figure out the direct/indirect band gap of 1D and 0D hybrid perovskites. It seems that 0D case shows an indirect bandgap if taking a close look at band edge (i.e., from Y to Z), which is different from most of 0D hybrid or inorganic perovskite calculated at the some theoretical level. In this case, the authors should consider the spin-orbit coupling (SOC) effects as well as using hybrid functional (to correct the underestimated band gap

raised by SOC) in their electronic bands calculations.

5) To understand the MHB effects and broadband emissions of 1D and 0D hybrid perovskite, the authors are encouraged to carry out the DFT calculations to demonstrate hydrogen bond strength, the position of self-trapping states, as well as the self-trapping exciton features in 1D and 0D hybrid perovskites.

6) In my opinion, it is misleading to say that " $\text{CH}_3\text{NH}_3\text{H}_1\text{I}_2+\text{H}_2\text{O}$ " functions as "molecular scissors" to conduct dimensionality reduction from 1D to 0D since the two crystals have the similar synthesis methods but only with the different precursor ratios. Did I miss something here? Otherwise, the authors should clarify this issue.

17-Aug-2019

Manuscript ID: NCOMMS-19-16936

Title: Collective hydrogen bonding isolates lead octahedra for white-emission improvement

Author(s): Cui, Bin-Bin; Han, Ying; Sun, Mingzi; Zhao, Yizhou; Liu, Lang; Cao, Guangyue; Du, Qin; Liu, Na; Zou, Wei; Wang, Lin; Wang, Jianpu; Huang, Bolong; Zhou, Huanping; Chen, Qi

Dear Editor:

We have thoroughly revised our manuscript and supporting information according to the reviewers' constructive comments. Following is our point-by-point response to these concerns. Main revisions and track changes are highlighted in red color in this uploaded documents for review only.

Reviewer #1 (Remarks to the Author):

Cui et al. report synthesis and characterization of both 1D and 0D lead-based perovskite-like crystals. The authors identify multiple-hydrogen-bonding as a reasoning behind the creation of a larger molecular spacer, which separates individual $[\text{PbBr}_6]^{4-}$ octahedron constructing 0D crystals exhibiting broadband warm white emission with 5 times higher quantum efficiency compared to the 1D counterpart. Finally, the authors use their data to confirm that the green emission observed in the Cs_4PbBr_6 is not from individual $[\text{PbBr}_6]^{4-}$.

The authors do report, to the best of my knowledge, the first 0D organic-inorganic lead-based crystal that will be of interest to many in the field. However, based on the comments below, I find the manuscript still requires critical improvements.

1. The quantum efficiency of this 0D crystals is below other reported white light lead-based 1D and 2D emitters (Zhou et al. *ACS Appl. Mater. Interfaces* 2017, 9, 40446-40451 and Gautier et al. *Adv. Mater.* 2019, 31, e1807383.).

Response: we thank the reviewer for the suggestive comments. Accordingly, we carefully studied the manuscript listed above regarding white light lead-based 1D and 2D emitters with high PLQEs. In the paper (*ACS Appl. Mater. Interfaces* 2017, 9, 40446), it observed efficient white emission of 1D perovskite crystals by doping Mn^{2+} ions. The improved PLQE of 28% was achieved in the Mn-doped 1D perovskites as compared to that of the intrinsic 1D perovskites (12%) with a blue emission. To be noted, the emission mechanism for Mn-doped crystals is different from that reported in our work, wherein self-trapped excitons are involved.

In the second paper cited above (*Adv. Mater.* 2019, 31, e1807383), the 2D emitters reported by Gautier *et al.* exhibit the crystal structure composed of *edge-sharing* PbBr_6 octahedra. In our reports, the 1D crystals are composed of the *corner-sharing* lead octahedra. As is generally accepted, the electronic band edge of the perovskite depends on the lead octahedra. And it is the overlap between the lead and halide atom orbitals that determines their emission properties ultimately. We actually provide an extra example showing the edge-sharing connecting mode results in higher PLQE than the corner-sharing mode in low dimensional halide perovskites. (page 2, 2nd paragraph, line 7) Our observation adds up to the knowledge pool that abounds people's understanding on this class of hybrid materials.

Here, we aim at a new type of hybrid materials with the isolated lead bromide octahedra that is never observed experimentally, which represents the novelty of the work. The PLQE is only one of its intrinsic properties, however

the significance of our finding does not lie on this. Our finding correlates the emission properties to the capability of structure deformation, which can be tailored by collective hydrogen bonding.

We understand the reviewer's concerns regarding the PLQE. We cited the two examples listed by the reviewer and discussed the issues in the corresponding paragraphs to further clarify the novelty and significance of our work. (page 2, 3rd paragraph, line 7; page 2, 2nd paragraph, line 7).

2. Lack of proper explanation regarding the origin of the broadband emission. In this regard, I recommend authors to perform transient absorption measurements.

Response: we appreciate the reviewer's comments and the transient absorption spectrum of the 0D crystals has been measured as suggested.

Figure R1 | (a) Transient absorption spectrum upon photoexcitation at 348 nm of 0D crystals. (b) Optical absorption spectra of the 0D bulk crystal

As is shown in Figure R1a (also Figure 3g in the revised manuscript), a broad pump-induced absorption with the peak at 420 nm (2.95 eV, a bit lower energy than bandgap) was observed for 0D crystals starting at 287 ps (excited at 348 nm with pulse energy of 14 $\mu\text{J}/\text{cm}^2$), which is likely assigned to self-trapped excitons. (STE, *Chem. Res.* 2018, 51, 619-627.) This is in contrast to the reported TA spectra of two-dimensional and three-dimensional lead-iodide hybrid perovskites, wherein narrow PL emission were observed with below-exciton bleaching features due to permanent trap states (*J. Am. Chem. Soc.* 2015, 137, 2089-2096; *Acc. Chem. Res.* 2018, 51, 619-627).

Furthermore, the sample did not show absorption near 580 nm often stemmed from the mid-band gap states of intrinsic defects (Figure R1b). It further suggests the instantaneous self-trapped exciton states with higher energy levels involved in the excitation process. (*ACS Energy Lett.* 2018, 3, 54-62.) Upon photoexcitation, electrons are excited to free-exciton excited states, which can undergo fast relaxation to self-trapped states, the multiple self-trapped states with different energies, causing a white broadband emission. It is different from the absorption spectrum of Cs_4PbBr_6 with green emission has an extrapolated absorption edge near 538 nm (2.30 eV) possibly due to the mid-gap states. (*ACS Energy Lett.* 2016, 1, 840-845; *J. Mater. Chem. C* 2016, 4, 10646-10653)

In our case, the emission from cm-sized single crystals and ball-milled powder are identical (Supplementary Figure 5b), indicating that the broad white-light emission of our 0D crystals is not from surface defects. In addition, as the intensity of the broadband emission exhibits a linear dependence on the excitation intensity (Supplementary Figure 4c), suggesting that emission does not arise from permanent defects.

We reorganized the figures in the revised manuscript to accommodate the TA results, corresponding discussions were also appended. (page 8, 1st paragraph, line 13)

3. The argument related to the debatable green emission observed in Cs_4PbBr_6 might not be valid. If isolated or individual $[\text{PbBr}_6]^{4-}$ should exhibit broadband emission and not narrow green emission, why is that broadband emission not observed even in the Cs_4PbBr_6 that do not show green emission? (Zhang et al. Cryst. Growth Des. 2018, 18, 6393-6398 and Akkerman et al. Nano Lett. 2017, 17, 1924-1930).

Response: Indeed, the green emission from Cs_4PbBr_6 is still debatable. Our findings are not trying to give a conclusion on this debate, but some experimental results for further discussion on this issue. Our observation mainly emphasizes the emission properties of individual lead bromide octahedra, and the underlying mechanism involves the STE states, which is supported by PL, TRPL, TA, and DFT simulation.

Figure R2 | Structural characterization of bulk 1D and 0D lead bromide hybrids. Structures of (a) 0D Cs_4PbBr_6 (b) 0D $(\text{C}_3\text{N}_3\text{H}_{11}\text{O})_2\text{PbBr}_6 \cdot 4\text{H}_2\text{O}$

It is meaningful to compare the Cs_4PbBr_6 to our materials system in this regard. The major structure difference lies on the cations, either Cs^+ or organic cations. The formation of self-trapped exciton is related to strong electron–phonon coupling mostly due to the less rigid crystal structure. As is shown in Figure R2a, the direct bond between Cs-Br greatly enhances the structural rigidity of all-inorganic Cs_4PbBr_6 0D perovskite. Owing to the soft nature of organic cations with larger size however, the extent for deformation of $[\text{PbBr}_6]^{4-}$ octahedra is more severe in our materials system upon excitation. It seems the Cs_4PbBr_6 is more rigid than our 0D crystals, wherein lead bromide octahedra is isolated by organic cations and water “cage”. As such, it is likely that their electron configuration at the band edge is different. Both two reasons may explain why the broadband emission from self-trapped states is not observed even in the Cs_4PbBr_6 .

It is therefore reasonable to expect a completely different photophysics due to the different electronic configurations at the band edge and the different structure deformability in this system. The scope of our work is not to clarify the origin of green emission in Cs_4PbBr_6 crystals tough, we appreciate the reviewer’s comments and modify the corresponding statements with additional discussions to avoid the potential misleading in the new manuscript (page 11, 2nd paragraph, line 1).

4. The figures quality is extremely poor except for Figure 4.

Response: we apologize for any inconvenience, and upload new manuscript with suitable format to keep the high quality of images.

Further comments:

1. Page 2 Line 4: identify what “A” site is.

Response: actually, “A” sites here are the locations of cations in 3D perovskite with the general formula of ABX_3 . This presentation is not suitable for 0~2D perovskites and the sentence is replaced by “compatible cations are introduced into the space between metal-halide octahedra to tune their connecting modes, which results in the variety in crystal structures and emission properties in this family of materials”. (page 2, 1st paragraph, line 6)

2. Scheme 1a: the 65% under the arrow is confusing and should be moved similar to Scheme 1b.

Response: revision has been made as suggested.

3. Figure 1: it is very difficult to identify the H_2O molecules.

Response: revision has been made as suggested. H_2O molecules are specifically marked in Figure 1.

4. Page 4 Line 45: “ $C_3H_9N_3O \cdot 2HBr$ is characterized by proton nuclear magnetic resonance (1H NMR) (Supplementary Figure 1)” Authors should provide some interpretation for the results and not simple report the spectra.

Response: revision has been made as suggested. 1H NMR (DMSO- d_6): δ 2.80 (q, $J = 6.0$ Hz, 2 H, $CH_2NH_3^+$), 3.20 (t, $J = 6.0$ Hz, 2 H, $CONHCH_2$), 5.72 (s, 3 H, $CONH_3^+$), 6.22 (s, 1 H, $CONHCH_2$), 7.71 (s, 3 H, $^+NH_3$). Therefore, two acid H^+ are belong to the two terminal amino in $C_3H_9N_3O \cdot 2HBr$, respectively. (page 5, 1st paragraph, line 4)

5. Figure 2c: please provide the molecules with numbering of the corresponding atoms or functional groups. What does the y-axis represent in this figure? Importantly, it seems that the carbonyl group (5) exist in the 0D crystal.

Response: revision has been made as suggested. We have improved Figure 2b, d to express the IR more clearly. The y-axis represent the normalized intensity of IR. The marks of carbonyl group (4) and carbonyl group (5) in Figure 2c are reversed and it has been corrected.

6. The 1D crystals appear to be yellowish in color and not colorless. Does that match with absorbance data?

Response: most bulk 1D crystals are colorless. The yellowish color is from the unclean teflon container and we have replaced the photo using a new one.

7. Authors should report PLE of both crystals.

Response: revision has been made as suggested. Spectrums on the left in Figure 3c are actually the PLE of crystals and the absorption data for the 0D crystal are showed in Supplementary Figure 4a.

8. Figure 3g: what are the “microscale crystals” in the legend? There is no discussion in the manuscript regarding this “microscale crystals”.

Response: the expression of “microscale crystals” here refers to ball-milled crystals. We’re confusing the two concepts and “ball-milled” will replace the “microscale” in Supplementary Figure 5a and b.

9. Authors should provide the fitting for the PL decay data.

Response: revision has been made as suggested.

10. Please check Figures numbers in SI.

Response: revision has been made as suggested.

11. Page 8 Line 182: “the major emission of 1D bulk”. Is there a minor origin of the emission?

Response: this is an expression mistake. There is no a minor emission of 1D bulk crystal and the “major” is removed in the new manuscript.

12. Would be very interesting if the authors could report the Cl and/or I counterparts.

Response: we appreciate the reviewer’s constructive suggestion. We have tried to grow crystals of Cl and/or I using the same methods. However, it’s difficult to obtain crystals based on Cl and the I counterpart but present another 1D structure which showed no emission property. Research on the Cl and/or I counterparts will continue and we hope report their meaningful functions in the future.

Reviewer #2 (Remarks to the Author):

The manuscript describes the synthesis, structural and optical characterization of a 0D and a 1D lead halide hybrid systems. In both cases, PbBr_6 perovskite octahedra are present, isolated in the 0D structure, and three corner connected in the 1D case, forming double infinite chains. Hydrogen bonding is observed, and is suggested as the driving interaction for the formation of these structures. Both phases exhibit white-emission upon UV excitation.

There are several issues that make the manuscript less appealing to a broader audience.

The authors are encouraged to seek publication elsewhere. For instance, Scientific Reports may be suitable.

Molecular Scissors: the term implies that the molecules described cut extended PbBr_3 blocks or sheets. However, no such proof is presented. The solution-based synthesis more likely produces the different structures due to charge transfer effects. The 1D structure, with three corners connected, reduces the need for charge balancing, while a larger concentration of organic cations is needed for the 0D structure.

Response: Thanks to the reviewer's valuable comments. We agree that "solution-based synthesis more likely produces the different structures due to charge transfer effects." However, different stoichiometry does not necessarily lead to different crystal structures. Especially in the hybrid perovskite system, the assembly of PbBr_3 blocks in different connecting modes depends on different processing parameters, among which the cations play an important role. For an example, butylammonium bromide (BA) is often employed to construct 2D perovskites. However, even we increase the stoichiometry to 1:4, 1:8, or higher, we do not obtain corresponding 1D/0D crystals. It means that charge transfer occurs in different conditions, wherein some specific molecules can tailor the final dimension of materials by extra effects to influence the connecting modes of PbBr_3 while others cannot. To visualize the effect of this type of molecules that can reduce the dimension of the PbBr_3 assembly, we employed "molecular scissors".

We understand the reviewer's concern that the ultimate driven force to create the hybrid crystal is the charge balancing. To avoid the potential misleading, we like to change the title to "*Collective hydrogen bonding isolates lead octahedra for white-emission improvement*", which emphasizes the collective hydrogen bonding effects between water and amides that create a larger spacer to isolate $[\text{PbBr}_6]^{4-}$ blocks.

2. In the 0D structure, small polaron behavior is expected, as in Cs_4PbBr_6 . Self-trapping will be similar, and a green emission is expected. In contrast, the presence of hydrogen bonding in the 0D structure likely produces a large local polarizability, with commensurate screening of the small polaron and different emission properties. Measurements of the dielectric constants of both the 0D and 1D phases would be helpful to address the question of the "green" emission for isolated PbBr_6 units. The "solid evidence" of characteristic broad band emission may need additional support.

Response: We appreciate the reviewer's constructive comments. It is true that the polaron behavior is critical to the emission behavior in the low dimensional hybrid systems. In order to measure the dielectric constant in 0D and 1D crystals, we fabricated the sheet samples with the thickness about 1 mm and diameter 1 cm by cold pressing, and dielectric measurements were performed in the frequency range 50 to 20 MHz. The real part of dielectric constants of 0D and 1D crystals are 11.75 and 11.27 at about 1 MHz, respectively, which are much smaller than the 3D halide perovskite (often over 50, *J. Phys. Chem. Lett.* 2011, 8, 4113-4121.; *J. Phys. Chem. C* 2018, 122, 13758-13766.) As compared to the 3D counterpart that produce large polarons, the excited electron/hole pairs are

closely confined within a metal-halide octahedron in the 0D crystal structure with the reduced dielectric constant. Therefore, the presence of hydrogen bonding in the 0D structure may not produce a large local polarizability, while small polarons are induced by the electron-phonon interactions and trap charge carriers, resulting in self-trapped excitons. (*Chem. Mater.* 2017, 29, 4129-4145). It helps to explain the broad-band white emission in 0D organic-inorganic hybrid in this work. We discussed the dielectric constant and polarons in the corresponding paragraph. (page 11, 2nd paragraph, line 2)

However, the luminescence mechanism of our 0D crystals and that of Cs₄PbBr₆ may be different. To further understand the underlying mechanism, we also performed the TA measurements, and the results are shown in Figure R1. Please also refer to Reviewer#1-Q2.

3. The DFT calculations are consistent with expectations, with the 0D phase showing a similar band gap as for Cs₄PbBr₆. How was the hydrogen bonding included in the electronic structure calculations?

Response: Thank you for the comment. For the accurate hydrogen bond calculation, it has been a challenge for both experiments and theoretical calculations for a long time (*Chem. Rev.* 2000, 100, 143; *Chem. Phys. Lett.* 1998, 290, 159). Neither computational nor experimental methods can easily decompose the individual contributions to the total energies. Presently, most reported hydrogen-bonding energies are typically based on dimerization energies, which include all of these interactions. Therefore, we have considered a novel approach to estimate the hydrogen bond strength based on the calculation of total electrostatic interactions and the monopole term of this interaction (*J. Phys. Chem. Lett.* 2017, 8, 6154-6159). By removing the monopole term of the electrostatic interactions between the cation and the anionic framework so that the strength of the dipolar interaction and higher-order terms forming the hydrogen bond can be determined. The number of the hydrogen bond is determined by the experimental results from Table 1. Therefore, the calculated hydrogen bond strength in 0D structure is listed in Table R1, which is also close to the previously reported hydrogen bond strength.

Table R1 | Hydrogen Bond Strength for 0D lead bromide hybrids

Structure	0 D
Number of Hydrogen bonds	8
Total Hydrogen-Bonding Energy (eV)	2.13
Hydrogen-Bonding Energy (eV)	0.27

4. The fact that the 1D structure contains linear Pb₂Br₉ moieties does not produce a noticeable dispersion along these moieties. The observed optical properties are thus expected to be similar to the 0D phase.

Response: we appreciate the reviewer's comments and agree this point. We observe the similar optical properties with respect to absorption and emission experimentally, which is in accordance with the DFT simulation results, just as the reviewer suggested.

Reviewer #3 (Remarks to the Author):

Chen and co-workers synthesized new 1D and 0D hybrid perovskite with broadband “warm” white emissions using the urea-amide cations. The authors found that the higher PLQE of 0D crystal than 1D case could be attributed to the multiple-hydrogen-bonding (MHB) effects which is reasonable explanation. The authors also carefully conducted several experiments to fully characterize the materials from few angles. Overall, the work is very interesting for the low-dimensional perovskite community; and it does indeed deserve publication in Nature Communications after revisions noted.

Response: We thank the reviewer for the positive comments.

1. The authors claimed that the higher PLQE of 0D crystal is attributed to the organic cations assembly and the reduction of the thermal vibrations. Does the excess H₂O also play a role in increasing the PLQE of 0D hybrid crystal?

Response: We appreciate the reviewer’s comments. Comparing to the 1D crystal, 0D crystals exhibit excessive H₂O in the crystal structure, which create larger spacers via hydrogen bonding with organic cations. These larger spacers isolate the PbBr₃ blocks leading to the luminescence from the individual inorganic octahedron. Except from the excessive H₂O in the crystal structure, we believe both 1D and 0D crystals may possess similar physio-adsorbed H₂O at the surface. This amount of water should be very small, since they were washed with acetone and vacuum-dried, which is a common trick used to remove additional water at surface during organic synthesis.

To answer the reviewer’s question, it is understood that additional water at surface may influence the emission properties of the materials dramatically. However, the 0D and 1D crystals experience the same treatment, so it is reasonable to expect the similar extent with respect to water adsorption at the crystal surface. The increase PLQE is mainly attributed to excessive H₂O in the crystal structure, which create larger spacers via hydrogen bonding with organic cations. Consequently, it further changes the exciton dynamics that influences the luminescence property of the system.

We tried to use higher temperature to remove the surface water. Unfortunately, the 0D crystals are not stable at high temperature (90 °C) to remove extra H₂O at the surface if exists. Please refer to Reviewer #3-Q2 for further information. Additional discussions are appended in the corresponding paragraphs to clarify the role of extra H₂O. (page 11, 1st paragraph, line 1)

2. Although the authors mentioned that “Both bulk 1D and 0D crystals retained emission color and intensity for at least one month” Does 0D crystal retain the same crystal structure at the high temperature after H₂O loss? And how about the PL spectra and PLQE of 0D crystal at high temperature after H₂O loss? This needs to be clarified in the revised version of the manuscript.

Response: We appreciate the reviewer’s constructive suggestion. Accordingly, we investigated the the H₂O loss in 0D crystal. After drying the 0D crystal under vacuum at 90°C for 24h, we found most 0D single crystals were changed in crystal structure due to H₂O loss, which was verified by PXRD. (Figure R3) In addition, PL spectrum (Figure R4, also Supplementary Figure 9) for the vacuum-dried 0D example shows the broadband emission with an additional peak of PbBr₂. It shows that the 0D crystals are not durable in high temperature. We clarify this point by additional discussions on page (page 11, 1st paragraph, line 5).

Figure R3 | Powder X-ray diffraction (PXRD) pattern of bulk 0D crystals after loss H₂O, as well as the simulated PXRD patterns of 0D and 1D single crystal structure.

Figure R4 | Emission spectra of bulk 0D crystals after loss H₂O.

3. I strongly recommend the authors to measure the exciton binding energy of the 0D structure and compared it to the inorganic 0D perovskite structure. This is also beneficial to take this new materials to the next level in terms of application.

Response: Thanks to the reviewer for his constructive comments. According to the literature (*ACS Energy Lett.* 2016, *1*, 840-845), we estimated the exciton binding energy by temperature-dependent PL, which is estimated to be 141 meV and 124 meV for 0D and 1D perovskites, respectively. (see Supplementary Figure 6) The exciton binding energy is higher than that in 3D perovskites, (*Chem. Phys.* 2014, *16*, 22476-22481; *Adv. Funct. Mater.* 2016, *26*, 2435-2445) but a bit lower than that in inorganic 0D perovskite structure (*Chem. Mater.* 2017, *29*, 7108-7113; *J. Mater. Chem. C* 2016, *4*, 10646-10653). It is in accordance with the reduced dielectric constant observed in the materials. (Please refer to Reviewer #2-Q2) Corresponding discussions regarding the binding energy is included in

the revised manuscript (page 8, 1st paragraph, line 25)

4. In Fig 4, it is difficult for the readers to figure out the direct/indirect band gap of 1D and 0D hybrid perovskites. It seems that 0D case shows an indirect bandgap if taking a close look at band edge (i.e., from Y to Z), which is different from most of 0D hybrid or inorganic perovskite calculated at the some theoretical level. In this case, the authors should consider the spin-orbit coupling (SOC) effects as well as using hybrid functional (to correct the underestimated band gap raised by SOC) in their electronic bands calculations.

Response: Thanks to the reviewer's comments. We have replotted the band structure of both 0D and 1D lead bromide hybrids systems with zoom in valence bands (Figure R5a and R5d, also revised manuscript Fig. 4a and 4b), in which the 0D system shows a direct bandgap of 3.74 eV, which is consistent with the previous reported 0D hybrid and inorganic perovskites. We also considered the norm-conserving treatment to the pseudopotential for comparison (Figure R5). Moreover, to address your concerns on the calculation method, we have conducted a detailed literature review to verify the reliability of the calculation methods. The estimated bandgap of the proposed material is highly closed to both reported experimental results and theoretical calculated band gap of Cs₄PbBr₆, indicating a reliable electronic result by the theoretical approach. In particular, for Pb-based perovskite materials, the consistency between experimental and calculated structural parameters is approved based on many previous works. Moreover, the electrostatic interactions as the main contribution to the interaction are well described by DFT-GGA. (*Adv.Mater.* 2019, *31*, 1900606; *Nano Lett.* 2017, *17*, 1924-1930; *J. Mater. Chem. A* 2013, *1*, 5628; *Phys. Rev. B* 2003, *67*,15540; *Chem. Phys. Lett.* 1999, *306*, 280-284.)

The consideration of SOC effect arises from the strong relativistic effect of Pb, which has been applied in previous work as well. However, ***it has been widely verified that the PBE functional well reproduces the experimental band gap due to a fortuitous error cancellation between the GGA underestimation and the overestimation by the lack of SOC effect.*** When the SOC is taken into account with PBE, the band structure retains while the conduction band degeneracies are changed, which largely reduces the bandgaps. Recently, Goddard *et al* (*Nano Lett.* 2016, *16*,3335-3340), showed the negligible difference between the calculated bandgap by PBE functional without SOC (2.22 eV) and the HSE + SOC method (2.21 eV). (*Nat Commun.* 2015, *6*, 7026 ; *ACS Energy Lett.* 2, 2017, 417-423; *Nano Lett.* 2016, *16*, 3335-3340; *J. Phys. Chem. Lett.* 2013, *41*, 2999-3005; *J. Phys. Chem. C* 2003, *117*, 13902).

Moreover, compared to the present calculation method, the hybrid functional with SOC requires much more time-cost, which cannot be accomplished in the limited revision time. Since the advantages in bandgap prediction by the hybrid functional with SOC will not substantially change the present conclusion, we believe that our calculations based on GGA+PBE functionals indeed supply reliable bandgap and band structures results.

Figure R5 | Density functional theory (DFT) calculations on band structures and photoluminescence mechanisms. (a, b) Calculated band structure of 0D and 1D lead bromide hybrids under ultrasoft pseudopotential. Inset: Zoom-in valence band (VB) structure. (c, d) Projected density of states of 0D and 1D lead bromide hybrids, respectively. (e, f) Calculated band structure of 0D and 1D lead bromide hybrids under norm conserving pseudopotential. Inset: Zoom-in valence band (VB) structure. (g, h) Real spatial contour plots for bonding and anti-bonding orbitals near E_F for 0D and 1D lead bromide hybrids, respectively.

5. To understand the MHB effects and broadband emissions of 1D and 0D hybrid perovskite, the authors are encouraged to carry out the DFT calculations to demonstrate hydrogen bond strength, the position of self-trapping states, as well as the self-trapping exciton features in 1D and 0D hybrid perovskites.

Response: Thank you for your kind suggestions. It has been a long-time challenge for DFT and experiments to conduct accurate characterizations for the strength of hydrogen bonds. (*Chem. Rev.* 2000, 100, 143 ; *Chem. Phys. Lett.* 1998, 290, 159). In particular, both computational and experimental methods cannot precisely decompose the individual contributions to the total energies among all the interactions. Thus, the hydrogen-bonding energies in previous reports are usually derived based on dimerization energies, which include all kind of interactions. Inspired by recent work by Butler et al. (*J. Phys. Chem. Lett.* 2017, 8, 6154-6159), we have determined the hydrogen bond strength based on the calculation of total electrostatic interactions and the monopole term. Therefore, the calculated hydrogen bond strength in 0D structure is listed in Table S1, which is also close to the reported hydrogen bond strength in lead bromide hybrids. (*J. Chem. Phys.* 2016, 144, 130901)

Table S1 | Hydrogen Bond Strength for 0D lead bromide hybrids

	0 D
Number of Hydrogen bonds	8
Total Hydrogen-Bonding Energy (eV)	2.13
Hydrogen-Bonding Energy (eV)	0.27

To date, transient absorption spectrum of the 0D crystals provides the most convincing evidence to confirm the self-trapping states under irradiation (page 3, column 1, 1st paragraph, line 10). Due to the lack of understanding of atomic arrangement of crystal in the self-trapping states, it is still difficult to carry out the DFT calculations to demonstrate the energy position of self-trapping states, as well as the self-trapping exciton features in 1D and 0D hybrid perovskites.

To support the luminescence is from the self-trapping states rather than other defects states, we calculate the point defects trap states on the electronic structures and defect formation energies using well-established methods. We focus on two vacancies (V_{Br} and V_{Pb}) because they are potentially dominant defects formed in perovskite materials. We find that the V_{Br} defect has a much smaller formation energy than V_{Pb} . Neither of transition levels of V_{Br} (3.0 eV) and V_{Pb} (4.2 eV) matches well with the broadband with emission at 570 nm (2.2 eV) of 0D structure. This further confirms our conclusion that the emission origin from the self-trapped states rather than intrinsic vacancy defects.

For these trap states, we have supplied a detailed discussion in the Supporting Information in the revised manuscript (Figure R6, also Supplementary Figure 6). In 0D and 1D systems, different electronic structures are noticed for vacancy states. In 0D, the Br-4p bands localized below the E_{F} while the anti-bonding orbitals of Pb-5p occupy 1 eV below the CB (Figure. R6a). Similarly, the Br-4p band is still pinned near the E_{F} in defective 0D system with Pb vacancy. However, the strong coupling between Pb-5p and organic spacer induces the elimination of anti-bonding states of Pb (Figure. R6b). In contrast, the organic spacer occupies a higher position than that of the Br-4p bands in different Br vacancy models. This indicates that the Br vacancy only contributes to the hole distribution near anti-bonding of Pb rather than involving into the modulation of trap states (Figure. R6c-d). Interestingly, we notice the well-matching bands between Br and organic spacer, which leads to the absence of hole states below CB (Figure. R6e). Finally, we compare the formation of vacancies in both 0D and 1D systems. Apparently, Br vacancies show smaller energy cost than Pb vacancies due to the coordination environment. The connected Br atom in $[\text{Pb}_2\text{Br}_9]$ unit also shows higher energy barrier in the formation than the normal Br atoms. The formation of Br vacancies is in the range of the excitation energy, supporting the potential contribution of Br vacancy (Figure. R6f)

Figure R6 | Density functional theory (DFT) calculations on intrinsic defects. (a, b) PDOS of Br and Pb vacancies in 0D lead bromide hybrids. (c-e) PDOS of Br and Pb vacancies in 1D lead bromide hybrids. (f) Calculated formation energies of vacancies in 0D and 1D lead bromide hybrids.

6. In my opinion, it is misleading to say that “ $CH_3NH_3H_{11}O^{2+}+H_2O$ ” functions as “molecular scissors” to conduct dimensionality reduction from 1D to 0D since the two crystals have the similar synthesis methods but only with the different precursor ratios. Did I miss something here? Otherwise, the authors should clarify this issue.

Response: We understand the reviewer’s concern. It is the stoichiometry difference that creates the different crystal structure. However, different stoichiometry does not necessarily lead to different crystal structures. For an example, butylammonium bromide (BA) is often employed to construct 2D perovskites. However, even we increase the stoichiometry to 1:4, 1:8, or higher, we do not obtain corresponding 1D/0D crystals. Comparing to BA, this molecule can alter the dimension by collective hydrogen bonding effects with different stoichiometry. To visualize

the effect of this type of molecules that can reduce the dimension of the PbBr₃ assembly, we give the name of “molecular scissors”.

We thank the reviewer for this comment. To avoid the potential misleading, we like to change the title to “*Collective hydrogen bonding isolates lead octahedra for white-emission improvement*”, which emphasizes the collective hydrogen bonding effects between water and amides that create a larger spacer to isolate lead bromide blocks. (Please also refer to Reviewer #2-Q1)

We hope that this revision is appropriate and the manuscript is suitable for publication now. However, should you have any more concerns, please feel free to contact us.

Sincerely,

Qi Chen, PhD, E-mail: qic@bit.edu.cn

Cui Bin-Bin, PhD, E-mail: cui-chem@bit.edu.cn

College of Materials Science and Engineering

Beijing Institute of Technology

Beijing, 100081, PR China

Reviewers' Comments:

Reviewer #1:

Remarks to the Author:

The authors have successfully replied to all comments/questions.

My concern is related to the comparison the authors made between their organic-inorganic 0D crystal and Cs₄PbBr₆ 0D crystal. Authors suggest that the absence of the broadband emission from self-trapped states in Cs₄PbBr₆ is basically due to the difference between Cs⁺ and organic cation resulting in an enhanced structural rigidity of all-inorganic Cs₄PbBr₆ 0D perovskite. If that is the case, then why the Sn counterpart (Cs₄SnBr₆) exhibits broad-band green–yellow PL at RT from STEs (Angew. Chem. Int. Ed. 2018, 57, 11329-11333).

I do not think that the authors provide a well articulated argument regarding the green emission from Cs₄PbBr₆ and I recommend it to be removed from the manuscript.

Reviewer #2:

Remarks to the Author:

The authors have addressed a number of the reviewer comments, and the manuscript is improved. However, the audience for this work is limited.

Furthermore, there are still issues that need to be addressed.

“Collective” hydrogen bonding: since the structure is periodic, the bonding is collective by definition. What particular “collective” action result from the hydrogen bonding?

The statement “ We hereby demonstrate a powerful toolbox to tune their connectivity by multiple-hydrogen-bonding effect...” is not proven. The authors have one example of water inclusion into their sample, with hydrogen bonding present. For a demonstration of a powerful toolbox, several examples would be expected.

The authors refer to “multiple hydrogen bonding effects”, however, such effects are not particularly well defined. The incorporation of water with hydrogen bonding to the amine groups in effect increases the size of the anion. Would a larger, different anion have the same effect?

The measured static dielectric constant of the order of 11 indicates that the excitons will be of the Frenkel type. A larger exciton binding energy would therefore be expected. If the organic molecules may also host excitons, have the authors checked for photoluminescence from the urea hydrobromide?

There is clearly an error: the authors claim that the CBM of the 1D system is dominated by Pb 4p bands. This is not possible.

The interaction between the lead bromide octahedra and the organic spacers, with a comparably small dielectric constant is expected to be small. It would therefore be prudent to talk about HOMO-LUMO, as “bands” imply dispersion and non-trivial charge carrier mobility. Excitons are then localized, and the interactions may result in charge-transfer excitons and higher excitonic states to produce a broadband emission. Since in both the 0D and 1D system, the HOMO-LUMO gap from the organic molecule is larger than the bonding-antibonding states difference from the PbBr₆ octahedra, where would the exciton self-trap?

Based on the DFT calculations, the isolated PbBr₆ octahedra do not interact strongly, the Br 4p

and Pb 5p derived bands are quite flat, indicating large effective mass.

There are a number of statements that should be either omitted, or strengthened:

“in 0D crystals, it may be owing to the MHB effects between the organic cations, the crystal structural stiffness is greatly improved, which leads to the reduction of the thermal vibrations, Consequently, defect states density within the electronic energy bandgaps may be reduced, and effect of electron-phonon coupling are alleviated possibly. Both effects are likely to contribute to reduce the non-radiative recombination channels, which is responsible for the enhanced PLQE.”

These statements are conjectures and speculations. It is not clear if increased stiffness from hydrogen bonding reduces electron-phonon coupling and thus, the self-trapping, or other mechanisms may explain the increased PLQE.

Reviewer #3:

Remarks to the Author:

The authors have addressed my major concerns. Therefore, I do recommend the current version of the manuscript for publication in Nature Communications.

Response to Reviewers' comments

Comments: blue color

Response: black color

Revised in the manuscript: red color

Reviewer #1(Remarks to the Author): The authors have successfully replied to all comments/questions. My concern is related to the comparison the authors made between their organic-inorganic 0D crystal and Cs₄PbBr₆ 0D crystal. Authors suggest that the absence of the broadband emission from self-trapped states in Cs₄PbBr₆ is basically due to the difference between Cs⁺ and organic cation resulting in an enhanced structural rigidity of all-inorganic Cs₄PbBr₆ 0D perovskite. If that is the case, then why the Sn counterpart (Cs₄SnBr₆) exhibits broad-band green–yellow PL at RT from STEs (*Angew. Chem. Int. Ed.* 2018, **57**, 11329-11333).

I do not think that the authors provide a well articulated argument regarding the green emission from Cs₄PbBr₆ and I recommend it to be removed from the manuscript.

Response: Thanks for the reviewer's kind suggestions. Indeed, it requires more efforts to fully understand the photo-physics in the organic-inorganic 0D crystals. We anticipate that the individual SnBr₆⁴⁻ octahedrons are different from the PbBr₆⁴⁻. In Cs₄PbBr₆, the strong direct bond between Cs-Br greatly enhanced structural rigidity. In addition, the distance between individual metal halogen octahedron is also an important factor to affect the formation of self-trapping states.

We quite understand the reviewer's concern. Since the experimental evidence of the Cs₄SnBr₆ single crystal structural characterization is not sufficient to compare the difference, we removed most arguments/speculations in the revised manuscript regarding the green emission from Cs₄PbBr₆ as the reviewer suggested. We only provide facts to the scientific community for further discussion on this issue in the future.

Reviewer #2 (Remarks to the Author): The authors have addressed a number of the reviewer comments, and the manuscript is improved. However, the audience for this work is limited. Furthermore, there are still issues that need to be addressed.

Response: We thank the reviewer for the recognition of our improvement in the revision. We also believe their constructive comments help to further improve our manuscripts. For the other concerns raised by the reviewer, we also supplied the corresponding explanation and revised the manuscript accordingly.

(1) “Collective” hydrogen bonding: since the structure is periodic, the bonding is collective by definition. What particular “collective” action result from the hydrogen bonding?

Response: We thank the reviewer for the suggestive comments. It is true that the crystal is made of a periodic arrangement of the atoms. However, by using “collective” here, we’d like to highlight the different hydrogen bonds that interact with each other within a short range of neighboring unit cells, rather than the typical periodical repetition occurred in the entire crystal (long-range). In this 0D material specifically, the abundant hydrogen bonding is playing a key role in three categories (**Figure R1a**) labelled in different colors: **1) water/amine group (blue), 2) water/carboxyl group (red), and 3) water/water (green).** Altogether we identify eight hydrogen bonds in **Table 1** and marked in **Figure R1b**. These hydrogen bonds link the adjacent organic molecules and water to form a larger stiff spacer. **We further identify their different functions when constructing the 0D crystals.**

(Please also refer to the response for *Reviewer#2 Question 2*) It is thus claimed that these local hydrogen bonds (within a short-range) *collectively* contribute to the isolation of the lead bromide octahedra to form the stable 0D crystal structure.

To avoid the potential misleading, we’d like to further change the title into “**Locally collective hydrogen bonding isolates lead octahedra for white-emission improvement**”. We further updated **Figure 2c** in the revised manuscript with **Figure R1a** and extended the corresponding discussions to articulate the different hydrogen bonding

and their collective effects in the formation of 0D perovskite. (page 6, 2nd paragraph, line 3)

Figure R1 | a) View of hydrogen-bonding in organic cations and water of the bulk 0D lead bromide crystal. b) The hydrogen bonding distances between water and organic cations of the bulk 0D lead bromide crystal (The [PbBr₆⁴⁻] octahedrons were eliminated for a better observation)

Table1 | Hydrogen Bond Parameters for 0D lead bromide hybrids.

D-H...A	D...A (Å)	angle at H (deg)
H-O-H...O=C	2.239	100.478
	2.964	107.777
N-H...O-H	2.347	123.268
N-H ₃ ...O-H	2.097	164.159
	2.812	110.452
	2.934	162.490
	2.982	126.064
H-O-H...O-H	2.172	149.310

(2) The statement “We hereby demonstrate a powerful toolbox to tune their connectivity by multiple-hydrogen-bonding effect...” is not proven. The authors have one example of water inclusion into their sample, with hydrogen bonding present. For a demonstration of a powerful toolbox, several examples would be expected.

The authors refer to “multiple hydrogen bonding effects”, however, such effects are not particularly well defined. The incorporation of water with hydrogen bonding to

the amine groups in effect increases the size of the anion. Would a larger, different anion have the same effect?

Response: Thanks to the reviewer's valuable comments, which inspire us to explore the significance of our work in a more explicit manner. To be noted, it is **the first time** to synthesize the organic-inorganic 0D hybrid perovskites-like materials system based on isolated PbBr_6^{4-} octahedra. To our surprise, the optical properties of the synthesized hybrid perovskites are completely different from that of the all inorganic Cs_4PbBr_6 0D single crystal, which has been widely studied at present.

Here we like to emphasize, the 0D organic-inorganic perovskites (especially lead-based) only receive limited success simply by adopting a large cation spacer in the synthesis approach. Prior to our work, many larger size cations with different structures have been reported to synthesize low dimensional lead halide materials. (*J. Am. Chem. Soc.* 2019, 141, 1171-1190) Interestingly, most of the resultant crystals are still limited to 2D layered structures (**Figure R2**), indicating large cation size are **not sufficient** criteria that lead to 0D crystals as we intuitively expected.

Figure R2 | Reported large organic cations that form corner-sharing (100)-oriented, (110)-oriented or (111)-oriented 2D halide perovskite structures (*J. Am. Chem. Soc.* 2019, 141, 1171-1190).

It is worth mentioning that the beauty of this work is to employ the unique cation urea-amide (N-(Aminocarbonyl)-1, 2-diaminoethane) (shown **Figure R3a**), which successfully results in the synthesis of the first 0D lead perovskite single crystal that ever reported. When we carefully examine the molecular structure of this cation, it

possesses the primary amine group (①), the secondary amide group (②), and the primary amide group -CO-NH_3 (③), which are essential to construct the 0D crystal. The amine/amide groups (①&②) with the special spatial configuration effectively interacts with bromide atoms in lead octahedra via multiple hydrogen bonds (**Figure R3b**, highlighted in blue), resulting in a corrugated crystal structure of nearly 90° folding angle. To be noted, the employment of similar cations with only amine/amide groups (①&②) is not able to achieve the complete isolation of the individual lead octahedra. Thus, this only leads to 2D corrugated crystals (**Figure R3c-e**)(*J. Am. Chem. Soc.* 2017, 139, 5210–5215). To further cut out the octahedra, the amide group (③) at the end of the cation display crucial effect, which provides abundant sites to form *locally collective hydrogen bonding* with adjacent molecules. When increasing the concentration of the cation, stronger intermolecular hydrogen bonding forces with H_2O creates a large spacer to isolate the individual lead bromide octahedra to form a 0D structure (**Figure R1a**). Therefore, the local hydrogen bonds collectively contribute to the modulation of the connectivity between lead octahedra, determining the dimensionality of lead halide hybrids.

Figure R3 | a) The amine in this work. b) Local hydrogen bonding causes the unique bending of the corrugated 1D structure. c, d) The organic molecule similar to the amine of our 0D single crystal while formed into a 2D structure. e) Local hydrogen bonding causes the unique bending of the corrugated inorganic layers (*J. Am. Chem. Soc.* 2017, 139, 5210–5215).

We understand the reviewer's concerns and have modified the corresponding discussions to illustrate the roles of different hydrogen bonds in constructing the 0D crystals. (page 6, 2nd paragraph, line 2)

(3) The measured static dielectric constant of the order of 11 indicates that the excitons will be of the Frenkel type. A larger exciton binding energy would therefore be expected. If the organic molecules may also host excitons, have the authors checked for photoluminescence from the urea hydrobromide?

Response: Thanks to the reviewer's comments. As suggested, we have carefully checked the PL of the urea hydrobromide. As shown in **Figure R4**, the emission spectrum of $C_3H_9N_3O_2HBr$ salts powder lies in the blue region with the peak at around 393 nm, which is apparently different from the broadband emission of our 0D single crystal. In addition, the emission of the organic molecule changes significantly with different excitation energy (*Adv. Optical Mater.* 2018, 6, 1800751; *Chem. Sci.* 2015, 6, 7222). In striking contrast, the emission of our 0D crystals is independent of excitation wavelengths (**Figure R4**). It indicates the final emission in the 0D crystals did not originate from the PL of the organic molecules. We cannot exclude the possibility that the organic molecules host the excitons, but the PL results demonstrate that the organic molecules are not the dominant contribution for the broadband emission. In addition, the PL, TRPL and TA results further support the emission mechanism involves the self-trap excitons (STE). (Please further refer to the response for *Reviewer#2 Question 5*)

Figure R4 | PL of 0D single crystals are independent of different excitation wavelengths and the PL of organic molecules.

In addition, the measured dielectric constant of our 0D crystals is 11, which is similar to other reported 0D halide perovskite-like crystals (**Table 2**). The calculated static dielectric constants of $(C_4N_2H_{14}Br)_4SnBr_6$, $(C_4N_2H_{14}I)_4SnI_6$, and Cs_4PbBr_6 are 11.9, 7.7, and 7.7, respectively (*J. Mater. Chem. C.* 2018, 6, 6398-6405). The exciton binding energy is similar to inorganic 0D perovskite structures from previous works (*Chem. Mater.* 2017, 29, 7108–7113; *J. Mater. Chem. C.* 2016, 4, 10646—10653). We agree with the reviewer on the high exciton binding energy in this materials system. Thus, we compared our result with other materials with corresponding explanation on this point in the revised manuscript. (page 12, 2nd paragraph, line 6)

Table 2 | Parameter comparison between our 0D and reported Cs_4PbBr_6 .

	dielectric constant	exciton binding energy (meV)
0D (this work)	11	141
Cs_4PbBr_6	7.7	159 ± 18
$(C_4N_2H_{14}Br)_4SnBr_6$	11.9	N / A
$(C_4N_2H_{14}I)_4SnI_6$	7.7	N / A

(4) There is clearly an error: the authors claim that the CBM of the 1D system is dominated by Pb 4p bands. This is not possible.

The interaction between the lead bromide octahedra and the organic spacers, with a comparably small dielectric constant is expected to be small. It would therefore be prudent to talk about HOMO-LUMO, as “bands” imply dispersion and non-trivial charge carrier mobility.

Response: We thank the reviewer for the suggestive comments. As DFT simulation predicted, the CBM of the 1D system is dominated by Pb 5p. We agree with the reviewer that HOMO-LUMO is more suitable in this materials system, and correct the corresponding expression. “the CBM of the 1D system is dominated by Pb 5p”. We modified the manuscript to remove the inappropriate terminology of “bands”.

(5) Excitons are then localized, and the interactions may result in charge-transfer excitons and higher excitonic states to produce a broadband emission. Since in both the 0D and 1D system, the HOMO-LUMO gap from the organic molecule is larger than the bonding-antibonding states difference from the PbBr₆ octahedra, where would the exciton self-trap?

Response: We thank the reviewer for the constructive comments. It is interesting to compare the self-trap excitons (STE) states to other possible emission mechanisms. The STE mechanism proposed in low dimensional hybrid perovskites is different from conventional organometal (e.g. Iridium complex), involving metal to ligand charge transfer (MLCT) that transfer excitons from the metal to organic molecules to be localized.

Recently, the self-trapping exciton in hybrid perovskites is well recognized and carefully investigated among the community (*Acc. Chem. Res.* 2018, 51, 619–627). The excitons are trapped in inorganic building blocks rather than organic molecules, wherein the organic spacers are not photo-active to be involved during the emission. (*J. Am. Chem. Soc.* 2014, 136, 13154–13157). During the 1st round revision, we have provided solid evidence to confirm that the emission in the 0D crystals is involving self-trapped states, which is acknowledged by the other two reviewers. The most

convincing data is transient absorption. As is shown in **Figure R5a** (also **Figure 3g** in the manuscript), we clearly observed a broad pump-induced absorption with the peak at 420 nm (2.95 eV, a bit lower energy than bandgap) was observed for 0D crystals starting at 287 ps, which is likely assigned to self-trapped excitons. (*Angew. Chem.* 2019, 131, 1 – 6). This is in contrast to the reported TA spectra of two-dimensional and three-dimensional lead-iodide hybrid perovskites, wherein narrow PL emission was observed with below-exciton bleaching features possibly due to permanent trap states (*J. Am. Chem. Soc.* 2015, 137, 2089-2096; *Acc. Chem. Res.* 2018, 51, 619-627). Furthermore, the sample did not show obvious absorption within the bandgap, which is often stemmed from the mid-band gap states of intrinsic defects (**Figure R6b**). As observed in the absorption spectrum of Cs₄PbBr₆ with green emission, which has an extrapolated absorption edge near 538 nm (2.30 eV) possibly due to the mid-gap states. (*ACS Energy Lett.* 2016, 1, 840-845; *J. Mater. Chem. C* 2016, 4, 10646-10653) By excluding those possible mechanisms, our result further suggests the instantaneous self-trapped exciton states with higher energy levels involved in the excitation process. (*ACS Energy Lett.* 2018, 3, 54-62.) Upon photoexcitation, electrons are excited to generate exciton states, which can undergo fast relaxation to self-trapped states. The multiple self-trapped states are expected to host different energy states, causing a white broadband emission. (*ACS Energy Lett.* 2018, 3, 54–62)

Due to the fast relaxation of the STE and limited techniques, it is not easy to capture the crystallographic information to describe the STE in an atomistic scale. However, the spectroscopic analysis supports the argument well. **To be noted, our manuscript is not the first report regarding the STE, which is supported by spectroscopic evidence that is generally accepted in this community.** We understand the reviewer's concerns, and we have strengthened the argument regarding the self-trap exciton in the revised manuscript by comparing other emission mechanisms. (page 9, 1st paragraph, line 9)

Figure R5 | (a) Transient absorption spectrum upon photoexcitation at 348 nm of 0D crystals. (b) Optical absorption spectra of the 0D bulk crystal

(6) Based on the DFT calculations, the isolated PbBr₆ octahedra do not interact strongly, the Br 4p and Pb 5p derived bands are quite flat, indicating large effective mass.

Response: Thanks for the comments. We appreciate the reviewer's comments and agree on this point. The presence of large spacer suppresses electronic coupling among (PbX₆)₄ octahedra as shown by the nearly dispersionless valence and conduction bands.

(7) There are a number of statements that should be either omitted, or strengthened: "in 0D crystals, it may be owing to the MHB effects between the organic cations, the crystal structural stiffness is greatly improved, which leads to the reduction of the thermal vibrations, Consequently, defect states density within the electronic energy bandgaps may be reduced, and effect of electron-phonon coupling are alleviated possibly. Both effects are likely to contribute to reduce the non-radiative recombination channels, which is responsible for the enhanced PLQE."

These statements are conjectures and speculations. It is not clear if increased stiffness from hydrogen bonding reduces electron-phonon coupling and thus, the self-trapping, or other mechanisms may explain the increased PLQE.

Response: Thanks for kind suggestions. As suggested by the reviewer, we have revised the corresponding discussions in the manuscript and removed those statements

that is not well supported by experimental results.

Mainly we refer to DFT simulation to explore possible mechanisms that may explain the increased PLQE. The corresponding discussions are appended in the revised manuscript (page 11, 2nd paragraph, line 8), also as quoted below.

Based on the DFT calculations, the isolated PbBr₆ octahedra do not interact strongly, the Br 4p and Pb 5p derived orbital energy levels are quite flat. The presence of large spacer suppresses electronic coupling among [PbX₆⁴⁻] octahedra as shown by the nearly dispersionless HOMO and LUMO levels. Therefore, no significant resonant transfer of the excitation energy is expected. Lowering the dimensionality of the inorganic networks from 3D, 2D, 1D, to 0D leads to increasingly more localized electronic states and consequently narrows the HOMO and LUMO distributing range, which promotes the self-trapping of excitons and stronger exciton emission. If the excitons are more localized, the probability of excitons getting trapped by intrinsic defects is significantly reduced, which decreases the non-radiative emission induced by the trapping near defects. Therefore, the resulting immobile excitons in our 0D have a low probability of interacting with the intrinsic defects (*J. Mater. Chem. C* 2018, 6, 6398-6405.), which further results in the increased PLQE as observed in the experiment.

Reviewer #3 (Remarks to the Author):

The authors have addressed my major concerns. Therefore, I do recommend the current version of the manuscript for publication in Nature Communications.

Response: Thank you for your recognition of our work. We highly appreciate the time and efforts you have paid for this manuscript as well as the valuable advice for helping us improve the manuscript.

Reviewers' Comments:

Reviewer #1:

Remarks to the Author:

The authors have addressed my major concerns and I recommend the manuscript for publication in Nature Communications.

Reviewer #2:

Remarks to the Author:

The authors have addressed most of the points raised previously.

The authors should consult a periodic table and check the labeling of the Pb bonding states in the figures and in the discussions in the text.

Reference 6 and 21 are identical.

The grammar needs to be checked.

Point-to-point response to reviewers' comments

Dear Reviewers,

Thank you for your precious time to constructive comments on our manuscript titled “**Locally collective hydrogen bonding isolates lead octahedra for white-emission improvement**” for *Nature Communications* (manuscript ID: NCOMMS-19-16936C). We sincerely appreciate your opinions and confirmation of our work. Accordingly, we have made the corresponding response and revisions based on your precious comments.

Reviewer #1 (Remarks to the Author):

The authors have addressed my major concerns and I recommend the manuscript for publication in *Nature Communications*.

Response: Thanks for your precious time and comments. We are grateful to your kind acceptance for this work, which is to be published in *Nature Communications*.

Reviewer #2 (Remarks to the Author):

The authors have addressed most of the points raised previously.

The authors should consult a periodic table and check the labeling of the Pb bonding states in the figures and in the discussions in the text.

Reference 6 and 21 are identical.

The grammar needs to be checked.

Response: Thank you for your precious comment, which help us to improve the manuscript significantly. Regarding the Pb valence orbital mistake, we are sorry about this typo due to our carelessness. It is true that the CBM of 1D and 0D crystals are mainly stemmed from Pb-6p. Fortunately, this typo will not affect the validity of our calculation results and the major conclusion. Accordingly, we have corrected this mistake in this revised manuscript thoroughly. Once again, we are sincerely sorry for the careless mistake. We further checked reference and grammar to correct most problems by our best efforts during the revision.